# Roles of *Candida albicans* Mig1 and Mig2 in glucose repression, pathogenicity traits, and *SNF1* essentiality

**Katherine Lagree**[1], **Carol A. Woolford**[1], **Manning Y. Huang**[1], **Gemma May**[1],
**C. Joel McManus**[1], **Norma V. Solis**[2,3], **Scott G. Filler**[2,3], **Aaron P. Mitchell**[1,4] *

**1** Department of Biological Sciences, Carnegie Mellon University, Pittsburgh, Pennsylvania, United States of America, **2** Division of Infectious Diseases, Los Angeles Biomedical Research Institute at Harbor-UCLA Medical Center, Torrance, California, United States of America, **3** Department of Medicine, David Geffen School of Medicine at UCLA, Los Angeles, California, United States of America, **4** Department of Microbiology, University of Georgia, Athens, Georgia, United States of America

* Aaron.Mitchell@uga.edu

**Data Availability Statement:** RNA-sequencing data are available at the SRA database (accession number PRJNA573104). All other data are within

## Abstract

Metabolic adaptation is linked to the ability of the opportunistic pathogen *Candida albicans* to colonize and cause infection in diverse host tissues. One way that *C. albicans* controls its metabolism is through the glucose repression pathway, where expression of alternative carbon source utilization genes is repressed in the presence of its preferred carbon source, glucose. Here we carry out genetic and gene expression studies that identify transcription factors Mig1 and Mig2 as mediators of glucose repression in *C. albicans*. The well-studied Mig1/2 orthologs ScMig1/2 mediate glucose repression in the yeast *Saccharomyces cerevisiae*; our data argue that *C. albicans* Mig1/2 function similarly as repressors of alternative carbon source utilization genes. However, Mig1/2 functions have several distinctive features in *C. albicans*. First, Mig1 and Mig2 have more co-equal roles in gene regulation than their *S. cerevisiae* orthologs. Second, Mig1 is regulated at the level of protein accumulation, more akin to ScMig2 than ScMig1. Third, Mig1 and Mig2 are together required for a unique aspect of *C. albicans* biology, the expression of several pathogenicity traits. Such Mig1/2-dependent traits include the abilities to form hyphae and biofilm, tolerance of cell wall inhibitors, and ability to damage macrophage-like cells and human endothelial cells. Finally, Mig1 is required for a puzzling feature of *C. albicans* biology that is not shared with *S. cerevisiae*: the essentiality of the Snf1 protein kinase, a central eukaryotic carbon metabolism regulator. Our results integrate Mig1 and Mig2 into the *C. albicans* glucose repression pathway and illuminate connections among carbon control, pathogenicity, and Snf1 essentiality.

## Author summary

All organisms tailor genetic programs to the available nutrients, such as sources of carbon. Here we define two key regulators of the genetic programs for carbon source utilization in the fungal pathogen *Candida albicans*. The two regulators have many shared roles, yet are

the manuscript and its Supporting Information files.

**Funding:** NIH/NIAID grants R01AI124566 (SGF, APM), R21AI144878 (APM) supported these studies. Website https://www.niaid.nih.gov/ The funders had no role in study design, data collection and analysis, decision to publish, or preparation of the manuscript.

**Competing interests:** The authors have declared that no competing interests exist.

partially specialized to control carbon acquisition and metabolism, respectively. In addition, the regulators together control traits associated with pathogenicity, an indication that carbon regulation is integrated into the pathogenicity program. Finally, the regulators help to explain a long-standing riddle—that the central carbon regulator Snf1 is essential for *C. albicans* viability.

## Introduction

Carbon metabolism is central to the growth and survival of all organisms. It provides both energy and biosynthetic building blocks. It is tightly controlled in most organisms to enable optimal use of diverse carbon sources. The ability to adapt to changing carbon sources is especially important for commensal and pathogenic microbes because microbial competitors and host factors can cause dynamic changes in the spectrum of carbon compounds available [1, 2].

Our focus is the fungus *Candida albicans*. It exists primarily as a commensal resident of the GI and GU tracts of humans and other warm-blooded animals. However, upon dysbiosis of the host environment it can cause infections that include oropharyngeal candidiasis, cutaneous candidiasis, vaginal candidiasis, and systematic or intra-abdominal candidiasis stemming from colonization of the patient's own GI tract [2, 3]. The ability of *C. albicans* to cause infection of diverse tissues and body sites depends upon its ability to regulate the utilization of diverse carbon sources [4].

Many of the mechanisms that govern carbon source utilization and regulation have been studied using the yeast *Saccharomyces cerevisiae* [5]. The extensive research from this model organism has been a useful guide for gene function analysis because genetic studies are more intractable in *C. albicans*. Additionally, this comparison is interesting because *C. albicans* is a human pathogen, so we might expect its regulation of metabolism and carbon source utilization to be different than in *S. cerevisiae*, a pathogen only on rare occasion. In fact, there are several unique features of *C. albicans* carbon regulation, such as distinctive transcriptional activators of glycolysis and alternative carbon source utilization [6, 7], loss of glucose-mediated catabolite inactivation [8], and loss of glucose-responsive post-translational modifications of the regulatory kinase, Snf1 [9]. Some differences in metabolic regulation have been directly linked to virulence [8], a connection that may inform new therapeutic strategies [4].

One form of metabolic regulation is called "glucose repression" or "carbon catabolite repression" [2, 4]. In the presence of glucose, a preferred carbon source, expression of genes for use of alternative (i.e., non-glucose) carbon sources is repressed. In *S. cerevisiae*, a central regulator of this pathway is the protein kinase *Sc*Snf1, also known as the AMP-activated protein kinase. *Sc*Snf1 is highly conserved among eukaryotes, and it functions to integrate diverse signals including metabolic and environmental changes like glucose restriction, oxidative stress, and alkaline pH [10].

*Sc*Snf1 is activated by the upstream protein kinase *Sc*Sak1 (<u>S</u>nf1 <u>A</u>ctivating <u>K</u>inase) and two paralogs, *Sc*Elm1 and *Sc*Tos1. *Sc*Snf1 is required for expression of glucose-repressed genes [11], a requirement that is mediated by the transcriptional repressor *Sc*Mig1 (<u>M</u>ulticopy <u>I</u>nhibitor of <u>G</u>AL gene expression). In response to glucose, *Sc*Snf1 is dephosphorylated, resulting in the activation of *Sc*Mig1 which represses expression of *Sc*Snf1-dependent genes [12, 13]. The paralog *Sc*Mig2 functions mainly to augment repression of a subset of the glucose-repressed genes controlled by *Sc*Mig1 [13, 14]. Although the activation of *Sc*Mig1 is directly controlled by *Sc*Snf1, *Sc*Mig2 has been shown to function independently of *Sc*Snf1 [13], is transcriptionally regulated by another pathway [15], and is post-translationally modified by ubiquitination

in response to carbon source [16]. In contrast, ScMig1 is regulated by carbon-responsive nuclear-cytoplasmic shuttling [17].

Elements of the Snf1 glucose repression pathway are conserved in *C. albicans* and in many other fungi. In *C. albicans*, Sak1 is required for utilization of alternative carbon sources, cell wall and membrane stress resistance, and expression of hexose transporter genes and glyoxylate cycle and gluconeogenesis genes [9]. Sak1 also influences production of long filamentous chains of cells called hyphae in a medium-dependent manner. *C. albicans* Snf1 clearly functions downstream of Sak1: it undergoes Sak1-dependent phosphorylation, and a strain that expresses a nonphosphorylatable Snf1-T208A mutant protein is unable to grow on alternative carbon sources [9]. Among downstream components, *C. albicans*, Mig1 is functionally conserved with *Sc*Mig1 as a repressor of several carbon utilization genes [18–20]. However, a *mig1Δ/Δ* mutation does not affect expression of most glucose-repressed genes [20]. Therefore, additional mediators of glucose repression in *C. albicans* have yet to be established.

One surprising feature of Snf1 function in *C. albicans* is that Snf1 is required for viability, unlike the situation in *S. cerevisiae* [9, 21–24]. Homozygous *snf1* null mutations have not been recovered through transformation or selection for mitotic recombinants [21, 22], methods that readily yield homozygous mutations in nonessential genes. *SNF1* essentiality has been most clearly established with a strain homozygous for a fusion of *SNF1* to a conditional promoter; the strain is able to grow only when the promoter is active [24]. Viable *snf1* mutations with altered function or expression [24–26] have permitted some phenotypic analysis, but it remains unclear whether the essential role of Snf1 depends upon its activity in the glucose repression pathway, or whether Snf1 has a novel second role that is essential [21].

Here we investigate the circuitry that drives the transcriptional response of *C. albicans* to glucose and the non-fermentable carbon source glycerol. Our findings reveal that Mig1 functions together with its paralog Mig2 to mediate glucose repression. Mig1 and Mig2 have both selective and shared functions in this glucose repression response. Each repressor also governs aspects of the *snf1Δ/Δ* and *sak1Δ/Δ* mutant phenotypes, thus tying them to the Snf1 pathway. Finally, our results argue that *SNF1* is essential because of exuberant repression by Mig1, thus connecting Snf1 essentiality to its role in the glucose repression pathway.

## Results

### Carbon control of gene expression

To investigate *C. albicans* gene expression response to carbon, we compared cells growing on the non-fermentable carbon source glycerol to those growing on glucose. RNA-sequencing (RNA-seq) was performed on wild-type *C. albicans* in YPG (Yeast Peptone Glycerol) and YPD (Yeast Peptone Dextrose) media grown to log phase at 37°C. A total of 489 genes were significantly altered (>2-fold change in RNA levels and $p < 0.05$) in YPG compared to YPD (S1 Table). Among those, 375 genes (77%) were up-regulated in YPG. These genes were enriched for processes that include organic acid catabolism, fatty acid catabolism, and hexose transport (ex. *CTN1*, *ICL1*, *MLS1*, *FOX2*, *HGT1*, *GAL1*; see S2 Table). The remaining 114 genes were down-regulated in YPG. These genes were enriched for processes that include carbohydrate metabolism and glycolysis (ex. *ENO1*, *GPM1*, *CDC19*, *TDH3*, *PFK1*, *PFK2*, *HXK2*; see S2 Table). Overall, the gene expression differences aligned well with known or expected features of cell metabolism.

We compared the differentially expressed genes in our YPG vs. YPD dataset to published gene expression datasets (Fig 1A and S3 Table). We focused on datasets in which both up- and down-regulated genes overlap with our dataset significantly ($p < 1E-07$; Fisher's Exact Test). We observed significant similarity to datasets intended to probe glucose repression, including

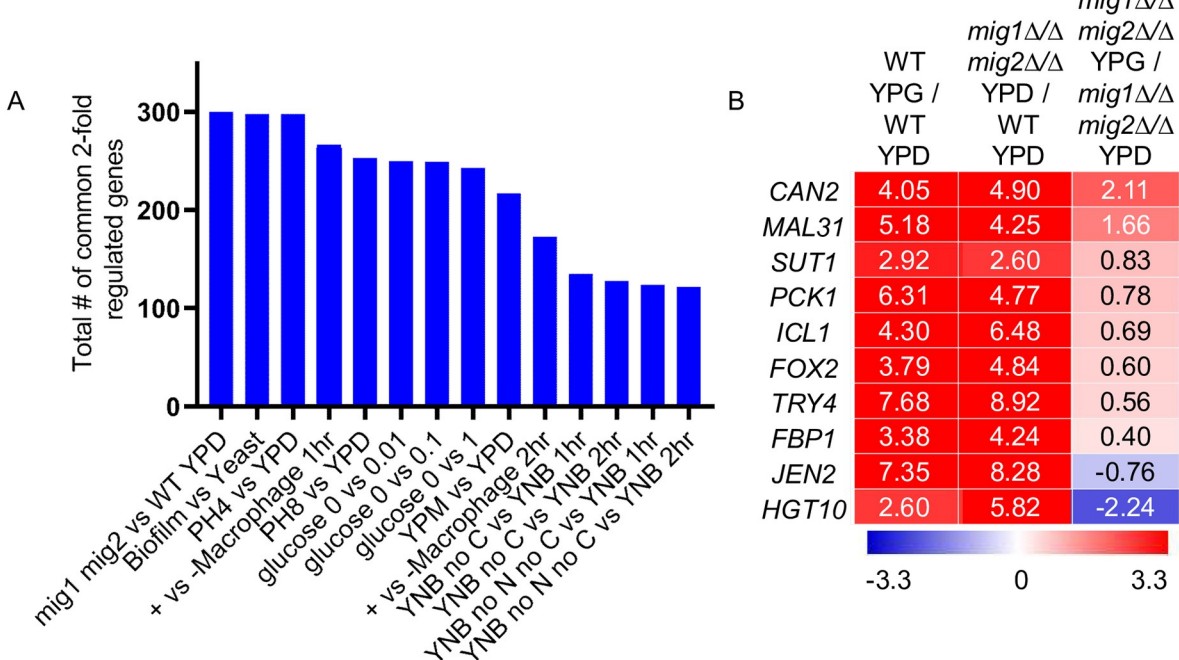

**Fig 1. Gene expression dataset comparisons.** Comparisons were performed using our genome-wide profiling data of wild-type *C. albicans* grown in YPG media compared to wild-type grown in YPD media **A**. Fisher's Exact Test depicting gene expression datasets most closely related to YPG gene expression. FET was used as previously described [31]. Up- or down-regulated genes with a 2-fold expression change cut-off were matched from 91 published expression datasets (S3 Table). **B**. Heatmap depicting gene expression of 10 metabolic genes chosen for comparison of glucose-repressed genes. Color scale corresponds to log2 fold change limits of 3.3 up (red) and 3.3 down (blue). Full gene expression datasets are available in S1 Table.

growth in YEP lactate vs YEP lactate + glucose (0.01, 0.1, or 1%), incubation in YNB medium lacking a carbon source or lacking a carbon source and nitrogen, and growth in YEP maltose vs YEP glucose [20, 27]. There was also significant similarity to *C. albicans* gene expression changes upon co-culture with macrophages [28], in keeping with current understanding that *C. albicans* adapts to alternative carbon sources in the macrophage phagosome [27, 28]. Some matching datasets were unexpected; for example, we detected significant similarity to cells in M199 pH 4 vs. YPD, and in M199 pH 8 vs. YPD [29]. The similarity in gene expression changes may reflect the difference in glucose concentration between M199 (0.1%) and YPD (2%). The greatest similarity detected was to an RNA-Seq comparison of biofilm cells vs. yeast cells [30], with 260 common up-regulated genes and 38 common down-regulated genes. The growth conditions for the biofilm study were Spider medium (for biofilm cells), with the carbon source mannitol, and SD +uridine (for yeast cells), with the carbon source glucose. Therefore, it is reasonable that glucose-repressed genes were expressed at higher levels in the biofilm cells than the yeast cells. (The *mig1Δ/Δ mig2Δ/Δ* double mutant vs. wild type comparison, shown in the figure, is discussed below). Similarities were driven by expression of a core set of 116 genes (S1 Fig; S1G Table). These dataset comparisons underscore the correlations between our results and glucose repression responses that have been measured previously through a variety of approaches.

## Functions of transcription factors Mig1 and Mig2 in gene expression

To define the determinants of *C. albicans* glucose-repressed gene expression, we investigated the transcription factors Mig1 and Mig2. Both Mig1 and Mig2 have homology to the *S.*

*cerevisiae* repressors *Sc*Mig1, *Sc*Mig2, and *Sc*Mig3 (S2 Fig), particularly in the DNA binding domain. Gene expression profiling has shown that Mig1 represses some carbon utilization genes in *C. albicans* [19, 20]. A recent study has implicated Mig2 in control of *C. albicans* cell size [32], but Mig2 function has otherwise been examined only in mutant screens.

To investigate the functions of Mig1 and Mig2, we created *mig1Δ/Δ*, *mig2Δ/Δ*, and *mig1Δ/Δ mig2Δ/Δ* double mutant strains using a transient CRISPR-Cas9 system [33]. These strains were then complemented using the CIP10 vector [34] to introduce one copy of a wild-type allele of *MIG1* or *MIG2* at the *RPS1* locus. For some experiments, we used a reconstituted strain in which a *mig1Δ* allele was replaced with a wild-type *MIG1* allele at the native locus [35] because the CIP10-*MIG1* plasmid yielded incomplete complementation (S4 Table). Substantial reversal of the mutant phenotypes in the complemented and reconstituted strains was verified through expression analysis for 28 genes (S4 Table).

To begin to define Mig1 and Mig2 functions, the wild-type, *mig1Δ/Δ*, *mig2Δ/Δ*, and *mig1Δ/Δ mig2Δ/Δ* double mutant strains were profiled using RNA-Seq in YPD medium. Additionally, the *mig1Δ/Δ mig2Δ/Δ* double mutant strain was profiled in YPG medium to compare its gene expression levels to the wild-type strain during growth using a non-fermentable carbon source. We found 488 genes that were significantly up-regulated ($p < 0.05$ and >2-fold change in RNA levels) in the *mig1Δ/Δ mig2Δ/Δ* double mutant compared to wild-type during growth in YPD, while only 144 genes were significantly down-regulated (Fig 2A, S1 Table). These profiling data suggest that Mig1 and Mig2 function mainly as transcriptional repressors.

To estimate direct regulatory interactions, we searched for the Mig1 binding motif SYGGRG [18] in the putative promoter regions of Mig1/2 target genes using PathoYeastract [36]. One or more Mig1 binding motifs were present in 86.4%, 93.6%, and 88.5% of genes that were up-regulated in *mig1Δ/Δ*, *mig2Δ/Δ*, *and mig1Δ/Δ mig2Δ/Δ* strains, respectively (Fig 2B). The overrepresentation of Mig1 binding motifs in the promoters of up-regulated genes in all three deletion mutants was statistically significant ($p < 0.05$) compared to promoters from all genes in the genome. Mig1 binding motifs were not enriched in the genes that were down-regulated in the mutant strains (Fig 2B), so these genes may respond indirectly to *mig* mutations. These results support the argument that Mig1 and Mig2 function as transcriptional repressors.

### Shared and selective targets of Mig1 and Mig2

The bulk of target genes are shared by both Mig1 and Mig2, but some target genes respond mainly to Mig1 or Mig2 (Fig 2A). These relationships are evident from both up- and down-regulated genes (Fig 2A). Because up-regulated genes are likely direct targets of the repressors, and because they have the most coherent functions, we focus on those genes below.

There were 67 genes that were significantly up-regulated in both the *mig1Δ/Δ* and *mig1Δ/Δ mig2Δ/Δ* mutant strains, but not in the *mig2Δ/Δ* mutant strain. We call these genes Mig1-selective targets. GO terms for this set of genes were enriched for transmembrane transporters of hexoses, with localization to the plasma membrane (S2 Table). Prominent among Mig1-selective targets were members of the *HGT* (high-affinity glucose transporters) family. The 20 *HGT* genes [37] encode transmembrane carbon transporters and at least one glucose sensor, Hgt4 [38]. Expression of these transporters is responsive to both the amount and type of carbon source available to the cells [37, 39]. Mig1 functions as the main repressor for six *HGT* genes and two predicted sugar transporters *HXT5* and *MAL31* in YPD (Fig 2C). This result is consistent with a previous RT-PCR experiment showing that Mig1 but not Mig2 represses *HGT1*,

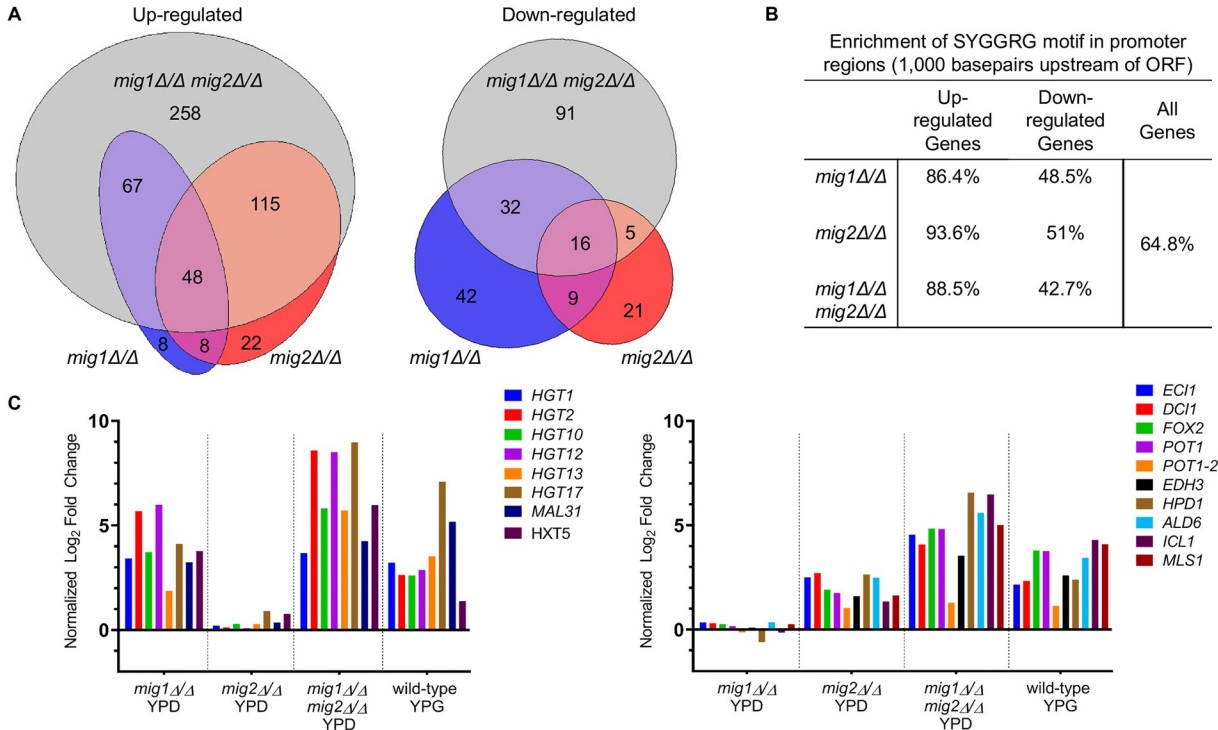

**Fig 2. Mig1/2 regulatory effects. A**. Venn diagram of genes regulated by Mig1, Mig2, or Mig1 and Mig2. Genes included in the diagram were significantly up- or down-regulated (*p*<0.05) by at least 2-fold compared to the wild-type strain (CW542) in YPD medium. **B**. Enrichment of SYGGRG motif in Mig1/2 target gene promoters. Promoter sequence was defined as 1,000 basepairs upstream of the open reading frame of up- or down-regulated genes. **C**. Fold change expression of chosen Mig1 and Mig2 selective genes compared to the wild-type strain analyzed by RNA-Seq and compared between datasets. Mig1 selective genes were significantly up-regulated in the mig1Δ/Δ and mig1Δ/Δ mig2Δ/Δ, but not the mig2Δ/Δ strain profiles. Mig2 selective genes were significantly up-regulated in the mig2Δ/Δ and mig1Δ/Δ mig2Δ/Δ, but not the mig1Δ/Δ strain profiles.

*HGT2*, and *HGT12* [39]. These results indicate that Mig1, and not Mig2, is the major repressor of 67 Mig1-selective target genes in glucose medium.

There were 115 genes that were significantly up-regulated in both the *mig2Δ/Δ* and *mig1Δ/Δ mig2Δ/Δ* mutant strains, but not in the *mig1Δ/Δ* mutant strain. We call these genes Mig2-selective targets. GO term enrichment includes fatty acid catabolic processes and lipid catabolic processes (S2 Table). Mig2-selective genes include the glyoxylate cycle genes *ICL1* and *MLS1* [40]; the *β*-oxidation genes *FOX2*, *POT1-2*, and *POT1*; and the genes *DCI1* and *ECI1*, which are predicted to encode enzymes involved in the *β*-oxidation of fatty acids (Fig 2C). We infer that Mig2, and not Mig1, is the major repressor of 115 Mig2-selective target genes in glucose medium. While Mig1 and Mig2 each repress genes involved in carbon utilization, there are functional distinctions among the Mig1- and Mig2-selective target genes.

The majority of the genes that are up-regulated in the *mig1Δ/Δ mig2Δ/Δ* double mutant are not up-regulated in either single mutant (258 genes; Fig 2A) indicating that Mig1 and Mig2 functions largely overlap. This profile is expected for genes where either Mig1 or Mig2 is sufficient for repression, and we refer to them as Mig1/2-shared genes. This gene set was enriched for metabolic genes with GO terms such as oxidation-reduction, transmembrane transport and lipid catabolic processes. Many of these genes encode enzymes associated with the peroxisome, similar to the Mig2-selective genes. For example, the peroxisome-related genes *PEX1*, *PEX2*, *PEX4*, *PEX6*, and *PEX12* were repressed only by Mig2, but *PEX3*, *PEX5*, *PEX8*, *PEX11*, *PEX13*, *PEX14*, and *PEX19* were repressed by either Mig1 or Mig2 (S1 Table). In addition, we

noticed that many Mig1- and Mig2-selective genes were derepressed further in the *mig1Δ/Δ mig2Δ/Δ* double mutant than in either single mutant. We infer for these genes that Mig1 or Mig2 is the major repressor, but that the other Mig1/2 protein can repress weakly in the absence of the first. Therefore, while Mig1 and Mig2 each have some specific target genes, for the most part they have redundant roles in gene regulation.

The relationship between Mig1/2-regulated genes and glucose-repressed genes can be seen in two dataset comparisons. The first comparison is our assessment of gene expression profile similarity (Fig1A and S3 Table). The differentially regulated gene set from our YPG vs. YPD samples showed greatest similarity to the differentially regulated gene set from our YPD-grown *mig1Δ/Δ mig2Δ/Δ* double mutant vs. wild type samples. Thus there is a strong correlation between genes repressed by Mig1 or Mig2 and those repressed by glucose. Second, we compared the gene expression levels of YPD- and YPG-grown wild-type and *mig1Δ/Δ mig2Δ/Δ* double mutant strains. We expected that, if Mig1 and Mig2 function primarily to exert glucose repression, then under derepressing conditions (YPG medium) the double mutation should have greatly diminished gene expression impact (Fig 1B). Indeed, for many genes, this prediction is fulfilled. Genes that are repressed over 8-fold by glucose in a wild-type strain (such as *ICL1*, *FOX2*, *TRY4*, and *JEN2*) are repressed less than 2-fold by glucose in the *mig1Δ/Δ mig2Δ/Δ* double mutant. These analyses support the conclusion that Mig1 and Mig2 are major mediators of glucose repression.

Though Mig1 and Mig2 repression activities are modulated by carbon source, our RNA-Seq profile data showed that *MIG1* and *MIG2* RNA levels are not (S1 Table). These results are consistent with prior studies of *C. albicans MIG1* [41] and distinct from the situation in *S. cerevisiae*, where carbon source affects *ScMIG1* and *ScMIG2* transcript levels [13, 15] as well as post-translational control [17, 42]. To determine whether Mig1 or Mig2 may be post-transcriptionally regulated, we constructed strains with HA epitope-tagged alleles. Mig1-HA was detectable on Western bots (see below); Mig2-HA was not. We examined Mig1-HA protein levels in cells grown in YPD, YPG, and Spider medium, which has the non-repressing carbon source mannitol. Mig1-HA protein levels were higher in YPD than in YPG or Spider medium, compared to a tubulin control (Fig 3A). In addition, Mig1-HA in Spider medium was detected in two

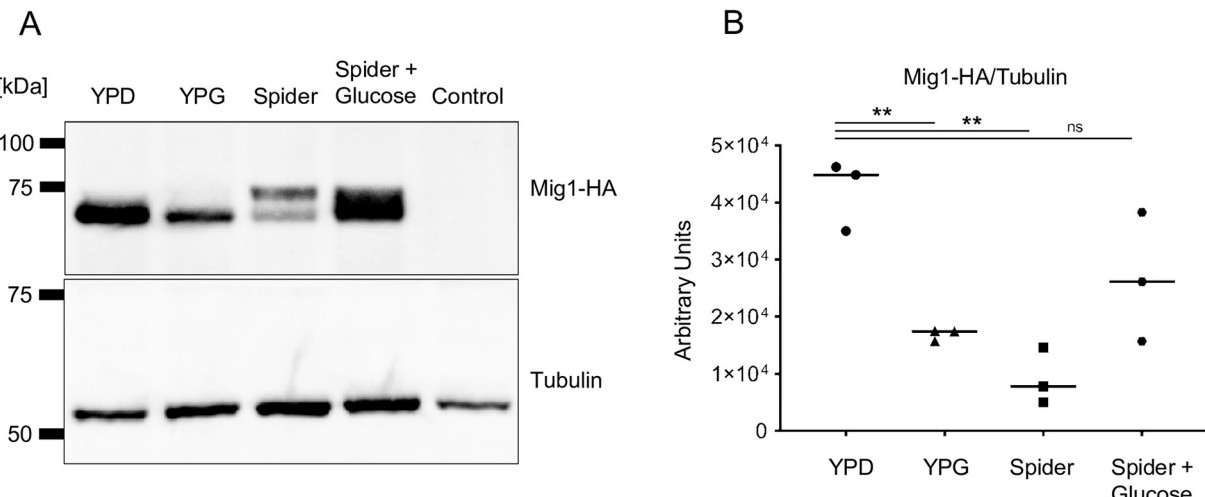

**Fig 3. Regulation of Mig1 protein accumulation in response to carbon source. A**. Western blotting was performed on cells grown in triplicate expressing HA-tagged Mig1 or the untagged wild-type strain (CW542) grown in YPD as a control. Soluble protein was extracted from strain KL1026 grown in YPD, YPG, Spider medium, and Spider medium with 2% glucose added 10 minutes before harvesting the cells. **B**. Densitometric analysis of Mig1 protein was performed using FIJI. Total HA-tagged signal was compared to total tubulin signal for loading control (Dunnett test **, *p*<0.01).

electrophoretic forms (Fig 3A). Mig1-HA levels were significantly different between YPD and YPG or Spider medium (Fig 3A and 3B and S3 Fig). In contrast, *MIG1* RNA levels were not significantly different between YPD and YPG or Spider medium (S1 and S5 Tables). To determine whether protein levels were responsive to glucose, we added 2% glucose to cells grown in Spider medium 10 minutes before harvesting. This addition of glucose resulted in a rapid increase in Mig1 protein levels (Fig 3A and 3B). All together, these results show that Mig1 protein levels are responsive to carbon source, while *MIG1* RNA levels are not. Regulation of Mig1 at the level of protein accumulation is more similar to regulation of *Sc*Mig2 than *Sc*Mig1 [17, 42].

### Impact of Mig1 and Mig2 on *sak1Δ/Δ* mutant growth

Sak1 is the main protein kinase that phosphorylates and activates Snf1 in *C. albicans* [9]. Because a *sak1Δ/Δ* is viable, we could use growth assays to test the hypothesis that Mig1 and Mig2 function downstream of Sak1 to control carbon physiology. To test this prediction, we examined *sak1Δ/Δ mig1Δ/Δ*, *sak1Δ/Δ mig2Δ/Δ*, and *sak1Δ/Δ mig1Δ/Δ mig2Δ/Δ* mutant strains, as well as control wild-type and *sak1Δ/Δ* strains. In agreement with prior studies [9], the *sak1Δ/Δ* mutant grew poorly on YPG and Spider media (Fig 4). The *sak1Δ/Δ mig1Δ/Δ* strain grew just as poorly as the *sak1Δ/Δ* mutant on these media, whereas the *sak1Δ/Δ mig2Δ/Δ* strain had improved growth on both media (Fig 4 and S4 Fig). The *sak1Δ/Δ mig1Δ/Δ mig2Δ/Δ* strain had further improved growth on both media. Reconstitution of a *SAK1* allele verified that the *sak1Δ/Δ* mutation was the cause of observed mutant growth defects (Fig 4). These growth tests

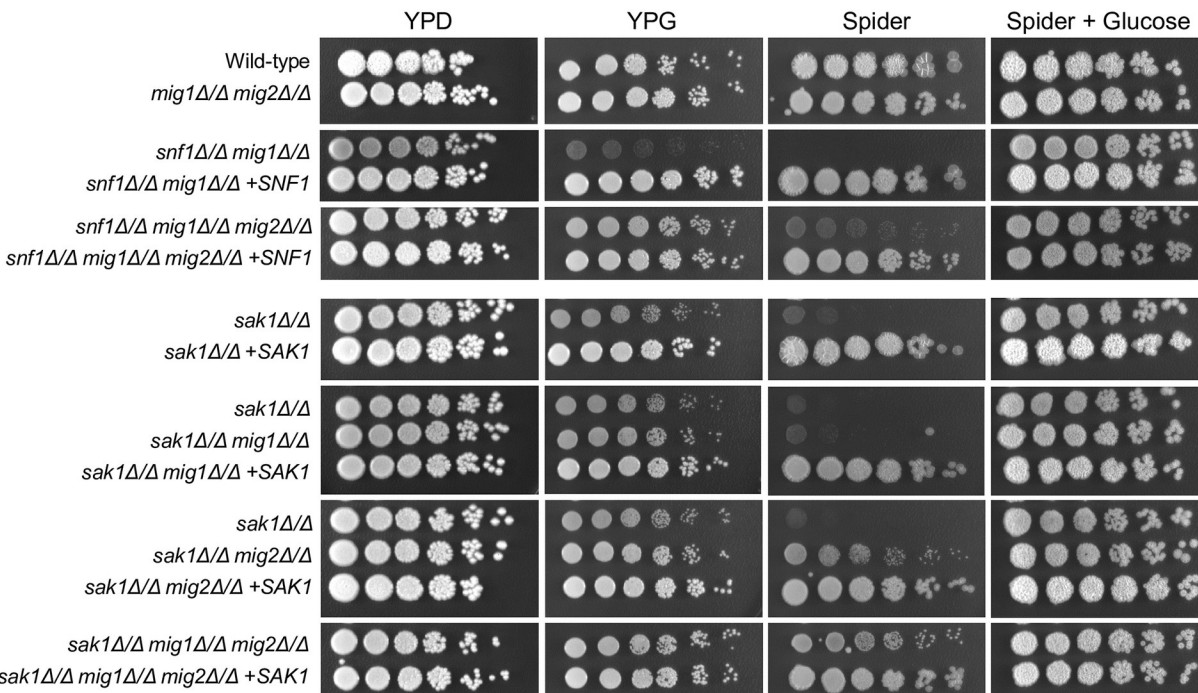

**Fig 4. Impact of *MIG1/2* mutations on growth of *sak1* or *snf1* mutant and validated strains.** Tenfold serial dilutions of strains KL988(*sak1Δ/Δ*), KL951(*sak1Δ/Δ mig1Δ/Δ*), KL960(*sak1Δ/Δ mig2Δ/Δ*), KL955(*sak1Δ/Δ mig1Δ/Δ mig2Δ/Δ*), KL992(*sak1Δ/Δ +SAK1*), KL972(*sak1Δ/Δ mig1Δ/Δ +SAK1*), KL990(*sak1Δ/Δ mig2Δ/Δ +SAK1*), KL974(*sak1Δ/Δ mig1Δ/Δ mig2Δ/Δ +SAK1*), and KL953 (*snf1Δ/Δ mig1Δ/Δ*), KL957(*snf1Δ/Δ mig1Δ/Δ mig2Δ/Δ*), KL970 (*snf1Δ/Δ mig1Δ/Δ +SNF1*), and KL976(*snf1Δ/Δ mig1Δ/Δ mig2Δ/Δ +SNF1*) were spotted on YPD (glucose), YPG (glycerol), Spider media (mannitol) and Spider media with 2% glucose replacing mannitol. Growth was visualized after 48 h of incubation at 37°C.

support the hypothesis that Mig1 and Mig2 act downstream of Sak1 to govern carbon physiology.

## Impact of Mig1 and Mig2 on *snf1Δ/Δ* mutant viability

*SNF1* is essential in *C. albicans* [9, 21–23], a possible consequence of essential genes that are targets of the glucose repression pathway or, alternatively, a glucose repression-independent role for Snf1 that is essential [21]. In *S. cerevisiae*, one major function of *Sc*Snf1 is to phosphorylate and inactivate *Sc*Mig1 [42, 43]. Therefore, we hypothesized that a *C. albicans snf1Δ/Δ* mutant may be inviable due to the inability to inactivate Mig1 or Mig2, leading to excessive repression of Mig1 or Mig2 target genes. Thus Snf1 essentiality would be tied to its function in the glucose repression pathway.

This hypothesis predicts that a *snf1Δ/Δ* mutant may be viable in a *mig1Δ/Δ*, *mig2Δ/Δ*, or *mig1Δ/Δ mig2Δ/Δ* mutant strain background. To test this hypothesis, we used the transient CRISPR approach to generate heterozygous and, potentially, homozygous *snf1Δ* mutants in each *mig* mutant strain. We recovered heterozygous *snf1Δ/SNF1* transformants in all three strains. In addition, we recovered homozygous *snf1Δ/Δ* transformants in both the *mig1Δ/Δ* and *mig1Δ/Δ mig2Δ/Δ* strains, though not in the *mig2Δ/Δ* strain. These observations support the hypothesis that a *snf1Δ/Δ* mutant is inviable due to repression of key target genes by Mig1.

Growth properties of representative *snf1Δ/Δ* derivatives of *mig1Δ/Δ* and *mig1Δ/Δ mig2Δ/Δ* strains suggested that loss of Mig1 does not bypass the need for Snf1 entirely. For example, *snf1Δ/Δ mig1Δ/Δ* colonies exhibited hyperfilamentation and yellow coloration, whereas *snf1Δ/Δ mig1Δ/Δ mig2Δ/Δ* colonies did not (S5 Fig). In addition, the *snf1Δ/Δ mig1Δ/Δ* strain displayed minimal growth on YPG or Spider media, whereas the *snf1Δ/Δ mig1Δ/Δ mig2Δ/Δ* strain displayed substantial growth on both (Fig 4). Poor Spider medium growth of the *snf1Δ/Δ mig1Δ/Δ* strain was relieved when glucose replaced mannitol as carbon source (Fig 4), so the Spider medium growth defect represents a carbon utilization defect. These observations support the hypothesis that Mig1 is the major repressor of genes that are required for *snf1Δ/Δ* viability, and that Mig2 contributes to repression of genes that enable more flexible *snf1Δ/Δ* carbon physiology. Our findings indicate that Snf1 is essential because of its function in the glucose repression pathway.

## Relationships among Mig1, Mig2, Sak1, and Snf1 in expression of glucose-repressed genes

To investigate how the phenotypic relationships among these regulators relate at the transcriptional level, we used 28 representative probes for Nanostring analysis from different metabolic pathways (i.e., glycolysis, gluconeogenesis, glyoxylate cycle, fatty-acid catabolism) and selected Mig1/2 target genes from our genome-wide analysis. RNA levels were assayed in a panel of single- and multi-gene mutants. *SAK1* or *SNF1* reconstituted strains served as controls for *sak1Δ/Δ* and *snf1Δ/Δ* mutants, and represented *mig1Δ/Δ*, *mig2Δ/Δ*, and *mig1Δ/Δ mig2Δ/Δ* mutant strains. RNA was isolated from cells grown in YPG medium, a condition in which we expected loss of Sak1 or Snf1 to cause reduced RNA levels from glucose-repressed genes.

We observed that 17 metabolic genes were significantly down-regulated (>2-fold and $p < 0.05$, compared to wild type) in the *sak1Δ/Δ* mutant (Fig 5 and S6 Table). Of those 17 genes, only three genes (*GAL10*, *JEN2*, *and PCK1*) were significantly down-regulated in the *sak1Δ/Δ mig1Δ/Δ* mutant, and six genes (*GAL10*, *JEN2*, *PCK1*, *HGT1*, *HGT2* and *MAL31*) were significantly down-regulated in the *sak1Δ/Δ mig2Δ/Δ* mutant (Fig 5 and S6 Table). In the *sak1Δ/Δ mig1Δ/Δ mig2Δ/Δ* triple mutant strain, only *GAL10* was significantly down-regulated. All other genes showed increased expression compared to the *sak1Δ/Δ* mutant, reaching levels

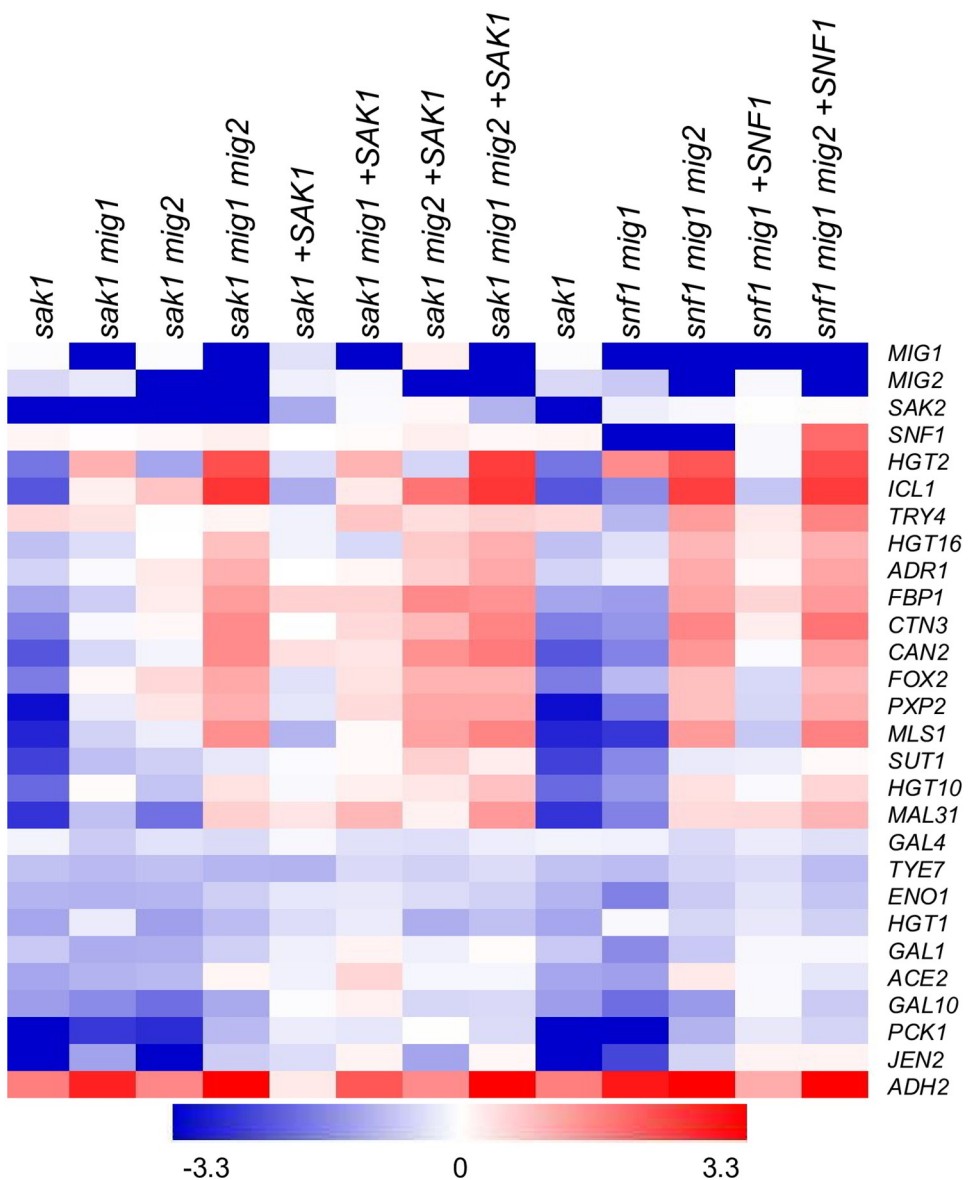

**Fig 5. Epistasis analysis among mutations in *MIG1, MIG2, SAK1*, and *SNF1*.** Heatmap of log2-fold changes in gene expression of selected carbon metabolism genes analyzed using Nanostring. RNA was extracted from strains CW542 (wild type), KL988(*sak1Δ/Δ*), KL951(*sak1Δ/Δ mig1Δ/Δ*), KL960(*sak1Δ/Δ mig2Δ/Δ*), KL955(*sak1Δ/Δ mig1Δ/Δ mig2Δ/Δ*), KL992(*sak1Δ/Δ +SAK1*), KL972(*sak1Δ/Δ mig1Δ/Δ +SAK1*), KL990(*sak1Δ/Δ mig2Δ/Δ +SAK1*), KL974(*sak1Δ/Δ mig1Δ/Δ mig2Δ/Δ +SAK1*), and KL953 (*snf1Δ/Δ mig1Δ/Δ*), KL957(*snf1Δ/Δ mig1Δ/Δ mig2Δ/Δ*), KL970 (*snf1Δ/Δ mig1Δ/Δ +SNF1*), and KL976(*snf1Δ/Δ mig1Δ/Δ mig2Δ/Δ +SNF1*) grown in triplicate in YPG medium at 37˚C. Fold change values were calculated by dividing expression of each gene to the wild type. The *sak1Δ/Δ* mutant strain analysis is duplicated for ease of comparison to the *snf1Δ/Δ* mutant strains because a *snf1Δ/Δ* mutant strain is not viable.

as high or higher than in the wild-type strain (Fig 5 and S6 Table). Therefore, the *mig1Δ* and *mig2Δ* mutations showed synergistic effects, reflected in the *sak1Δ/Δ mig1Δ/Δ mig2Δ/Δ* strain, just as they had in their effects on glucose-repressed genes in an otherwise wild-type background (Fig 2A). The fact that the reconstituted *sak1Δ/Δ mig1Δ/Δ mig2Δ/Δ +SAK1* strain and the triple mutant *sak1Δ/Δ mig1Δ/Δ mig2Δ/Δ* strain had nearly identical gene expression profiles (Fig 5) indicates that, for this panel of genes, Mig1 and Mig2 are responsible for almost all gene expression impact attributable to loss of Sak1. These results support the conclusion that

Mig1 and Mig2 mediate repression of Sak1 target genes. The synergistic effects of *mig1Δ* and *mig2Δ* mutations support the conclusion that Mig1 and Mig2 have partially overlapping roles in mediating repression.

The gene expression profile of a *snf1Δ/Δ* mutant cannot be assayed, so we used the *sak1Δ/Δ* strain for comparison to our *snf1Δ/Δ* strain panel (Fig 5 and S6 Table). Few differences were seen between the *sak1Δ/Δ* and *snf1Δ/Δ mig1Δ/Δ* strains except for the glucose transporter genes *HGT1* and *HGT2*. Inviability of *snf1Δ/Δ mig2Δ/Δ* strain prevented its analysis. The reconstituted *snf1Δ/Δ mig1Δ/Δ mig2Δ/Δ +SNF1* strain and the *snf1Δ/Δ mig1Δ/Δ mig2Δ/Δ* triple mutant strain had nearly identical gene expression profiles (Fig 5 and S6 Table). This observation supports the conclusion that Mig1 and Mig2 are responsible for almost all gene expression impact attributable to loss of Snf1. These results indicate that Mig1 and Mig2 both mediate repression of Snf1 target genes.

## Mig1 and Mig2 promote tolerance of cell wall inhibitors

Carbon source influences *C. albicans* stress resistance and cell wall composition [44, 45], so it seemed possible Mig1 and Mig2 may be required for tolerance of cell wall inhibitors. We tested this idea with growth assays (Fig 6A and 6B) in the presence of the cell wall inhibitor caspofungin. The *mig1Δ/Δ* mutant showed mild caspofungin hypersensitivity, as detected with a quantitative assay (Fig 6B). The *mig2Δ/Δ* strain was not caspofungin hypersensitive. The *mig1Δ/Δ*

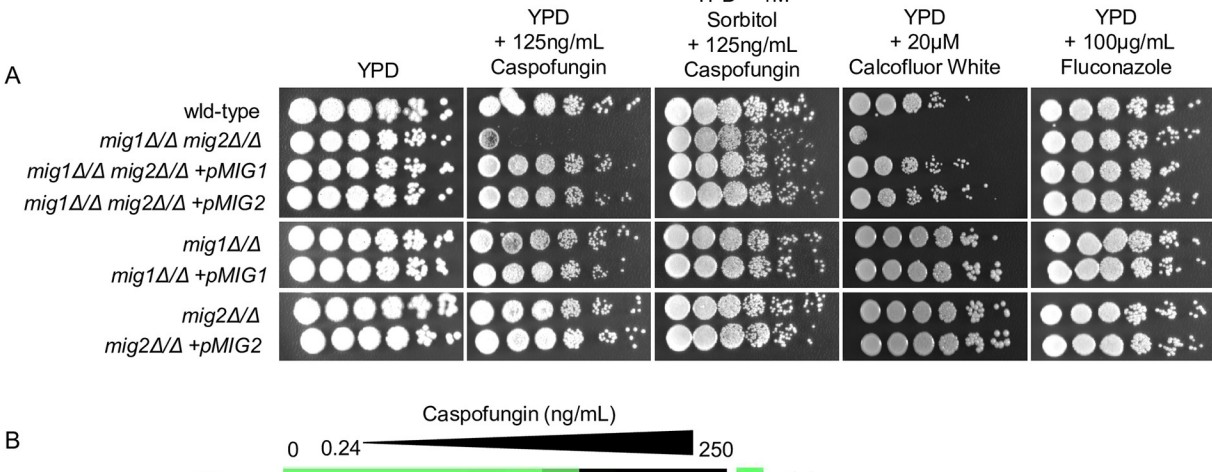

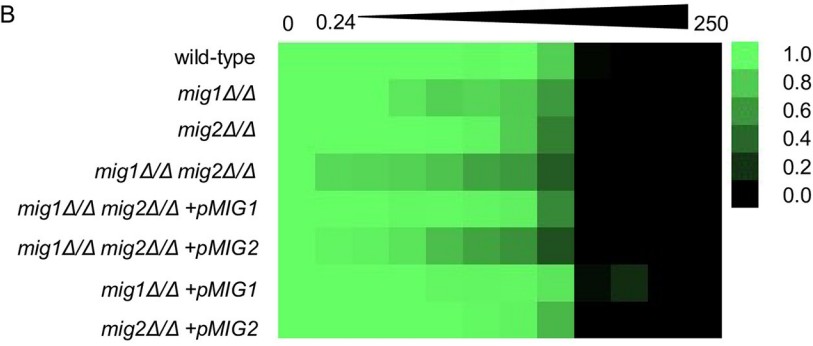

**Fig 6. Growth of Mig1/2 mutant and complemented strains during cell wall stress. A**. Plate dilution sensitivity assay. Tenfold serial dilutions of the indicated strains were spotted on YPD and YPD with 100 ng/mL Caspofungin, 100 ng/mL Caspofungin and 1 M Sorbitol, 20 μM Calcofluor White, and 100 μg/mL Fluconazole. Growth was visualized after 48 hours of incubation at 37°C. **B**. Quantitative caspofungin sensitivity assay in 96 well plates. Cells were incubated in liquid YPD media at 37°C containing 2-fold dilutions of Caspofungin. Absorbance was read at 600nm after 48 hours of incubation. Data was averaged from three biological replicates of duplicate measurements and normalized to cell growth without antifungal.

*mig2Δ/Δ* double mutant showed strong caspofungin hypersensitivity. Its hypersensitivity was partially relieved by complementation with one copy of *MIG2*, and essentially reversed by complementation with one copy of *MIG1* (Fig 6B). The double mutant was hypersensitive to another cell wall perturbing agent, calcofluor white, and its caspofungin hypersensitivity was relieved by the osmotic stabilizer sorbitol (Fig 6A). The *mig1Δ/Δ mig2Δ/Δ* double mutant was not hypersensitive to the ergosterol synthesis inhibitor fluconazole (Fig 6A). These results support the idea Mig1 and Mig2 are required for cell wall integrity or biogenesis, probably through impact on carbon physiology, rather than general stress tolerance.

## Mig1 and Mig2 promote filamentation and biofilm formation

Carbon source also affects one of the most prominent *C. albicans* virulence traits, the ability to form hyphae [46]. Therefore, we tested the mutant strains for defects in hyphal formation. Under our conditions, the *mig1Δ/Δ* mutant showed a modest but statistically significant decrease in hyphal length compared to wild type while the *mig2Δ/Δ* mutant showed no hyphal defect (Fig 7C). The *mig1Δ/Δ mig2Δ/Δ* double mutant had a pronounced defect in hyphal

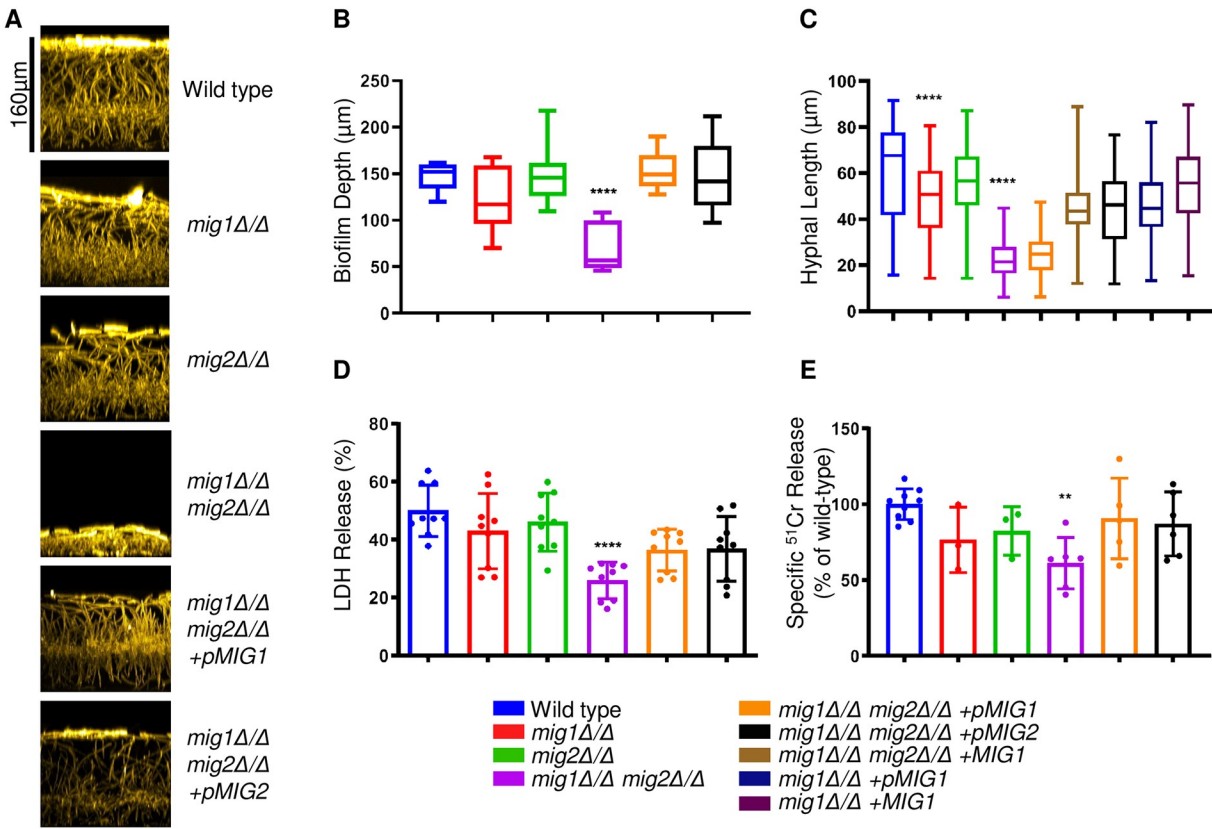

**Fig 7. Mig1/2 promote filamentation, biofilm formation, and affect host cell interactions *in vitro*. A**. Biofilms were grown for 24 h on silicone squares in RPMI media at 37˚C. Fixed biofilms were stained with Concanavalin A Alexa Fluor 594. Representative side-view projections were processed and pseudocolored using ImageJ. Scale bar corresponds to depth of the wild-type **B**. Biofilms were measured using confocal microscopy. Values shown are triplicate measurements from three biological replicates. ****, $p < 0.0001$. **C**. Indicated strains were grown for 4 h in RPMI at 37˚C. Fixed cells were imaged using DIC microscopy and a 20x objective. Hyphal lengths were measured from yeast cell to hyphal tip from >80 cells from 10 fields of view. ****, $p < 0.0001$ **D**. J774.1 macrophage cytotoxicity was measured by lactate dehydrogenase release (LDH) after 5 h of coincubation with *C. albicans* strains as indicated. Percentage of LDH release was calculated relative to max release wells containing lysis solution. Values shown are mean with SD from duplicate measurements of three biological replicates. ****, $p < 0.0001$. **E**. Human endothelial cell damage was assessed by $^{51}$Cr release following coincubation with *C. albicans* cells for 3 h. Percentage of Chromium release was calculated by comparison to release from wild-type (release–spontaneous/total incorporation–spontaneous)**, $p < 0.01$.

length compared to the wild type (Fig 7C). This defect was not reversed after integration of *MIG1* at the *RPS1* locus (strain labeled *mig1Δ/Δ mig2Δ/Δ + pMIG1)*, though it was reversed by integration of one copy of the *MIG1* at the native locus (strain labeled *mig1Δ/Δ mig2Δ/Δ + MIG1)*. The requirement for native locus integration may reflect regulatory sequences that were absent from the plasmid used to integrate at *RPS1*. Hyphal formation is required to form stable multicellular biofilm communities [47], and the double mutant showed a significant reduction in biofilm depth compared to the wild-type and single mutant strains (Fig 7A and 7B). These results indicate that Mig1 and Mig2 are required for normal hyphal morphogenesis and biofilm production.

## Mig1 and Mig2 affect host cell interaction *in vitro*

Metabolic plasticity is required for *C. albicans* to escape from macrophages *in vitro*. Following phagocytosis, *C. albicans* up-regulates alternative carbon utilization gene expression [27, 48] and undergoes a yeast-to-hyphal transition that disrupts the membrane and triggers macrophage cell death [49]. Because the *mig1Δ/Δ mig2Δ/Δ* mutant up-regulates alternative carbon utilization gene expression, it seemed possible that the mutant would have increased capacity to damage macrophages. On the other hand, because the *mig1Δ/Δ mig2Δ/Δ* mutant has a defect in hyphal formation, it seemed possible that it would have diminished capacity to damage macrophages. To determine which of these phenotypes may have overriding impact, we tested the interaction of *mig* mutants with J774.1 murine macrophages. Lactate dehydrogenase release was used as a readout for macrophage cell damage after incubation with wild-type, *mig1Δ/Δ*, *mig2Δ/Δ*, *mig1Δ/Δ mig2Δ/Δ*, and the relevant complemented strains. The double mutant strain showed significantly decreased cell damage capacity compared to the wild-type or single mutant strains (Fig 7D). Thus, despite a head start that the *mig1Δ/Δ mig2Δ/Δ* mutant strain may have in utilizing alternative carbon sources, other aspects of its phenotype—potentially the hyphal defect—cause decreased macrophage pathogenesis.

We also assayed host cell interaction through the ability of the mutant strains to cause endothelial cell damage [50], using primary human umbilical vein endothelial cells. The double mutant strain showed a significant defect in cell damage compared to the wild-type and single mutant strains (Fig 7E). These results confirm that Mig1 and Mig2 are redundant for major biological functions and indicate that Mig1 and Mig2 are required for pathogenicity-associated phenotypes.

## Discussion

Carbon metabolism and its regulation are central to the pathogenic capability of *C. albicans*. Carbon metabolic genes undergo dynamic expression changes in numerous colonization and infection models, and defects in carbon metabolism or its regulation disrupt the ability to colonize the GI tract, duel with macrophages and other host cells, and infect host tissues [2, 4]. Here we have extended our understanding of one major carbon regulatory pathway, the Snf1 pathway. We have identified transcription factors Mig1 and Mig2 as mediators of glucose repression in *C. albicans* and defined their functional activities and connections to upstream regulators Snf1 and Sak1. The Mig1/2 orthologs *Sc*Mig1/2 mediate glucose repression in *S. cerevisiae*, so in a broad context their functions are conserved in *C. albicans*. However, our studies reveal several distinctive features of Mig1/2 in *C. albicans*. First, Mig1 and Mig2 have relatively equivalent roles in gene regulation, whereas *Sc*Mig1 is the major mediator of glucose repression in *S. cerevisiae*. In addition, Mig1 is regulated at the level of protein accumulation, a difference from the regulation of *Sc*Mig1. A third unique feature is that Mig1 and Mig2 are together

required for several traits associated with pathogenicity, including formation of hyphae and biofilm, cell wall inhibitor tolerance, and damage of mammalian host cells. Lastly, Mig1 is required for the essentiality of the Snf1 protein kinase in *C. albicans*. Snf1 is not essential in *S. cerevisiae*, and the riddle of Snf1 essentiality has not been addressed to our knowledge in a model system. Below we discuss the functions of Mig1 and Mig2 and their roles in *C. albicans* biology.

## Mig1 and Mig2 function

Our results indicate that Mig1 and Mig2 largely overlap in function as repressors of glucose-repressed genes, yet each has some selective bias in target genes (summarized in Fig 8). We defined selective targets based on their significant up-regulation in one single mutant and not the other. Mig1-selective targets include carbon transporters; Mig2-selective targets include β-oxidation genes. This outcome was unexpected from work in *S. cerevisiae; Sc*Mig2 does not regulate specific glucose-repressed genes without *Sc*Mig1 [14]. In fact, we observed that Mig2 had more selective target genes than Mig1 (115 vs 67; Fig 2A). However, most selective target genes have further increased RNA levels in the *mig1Δ/Δ mig2Δ/Δ* double mutant compared to the *mig1Δ/Δ* or *mig2Δ/Δ* single mutants (Fig 2C). Therefore, Mig1 and Mig2 both contribute to repression of most selective target genes.

The majority of genes that are up-regulated in a *mig1Δ/Δ mig2Δ/Δ* double mutant strain show little expression change in either a *mig1Δ/Δ* or *mig2Δ/Δ* single mutant strain. These 258 genes (Fig 2A) represent targets that can be fully repressed by either Mig1 or Mig2. The large number of such genes argues that Mig1 and Mig2 most often function interchangeably.

We note that 228 genes are significantly up-regulated in a *mig1Δ/Δ mig2Δ/Δ* double mutant vs wild type in glucose but not in the wild type grown in glycerol vs glucose. This gene class might be taken to indicate that Mig1 and Mig2 have functions beyond a role in glucose repression. The 228 genes are enriched for functions in oxidation-reduction and peroxisome biogenesis. In yeast, peroxisomes function as sites for hydrogen peroxide reduction and fatty acid metabolism [51] and the abundance of these specialized organelles increases in response to non-glucose media such as oleic acid [52]. Therefore, some of the genes may have medium-specific glucose repression responses. Indeed, we find that 65 of the genes are up-regulated in

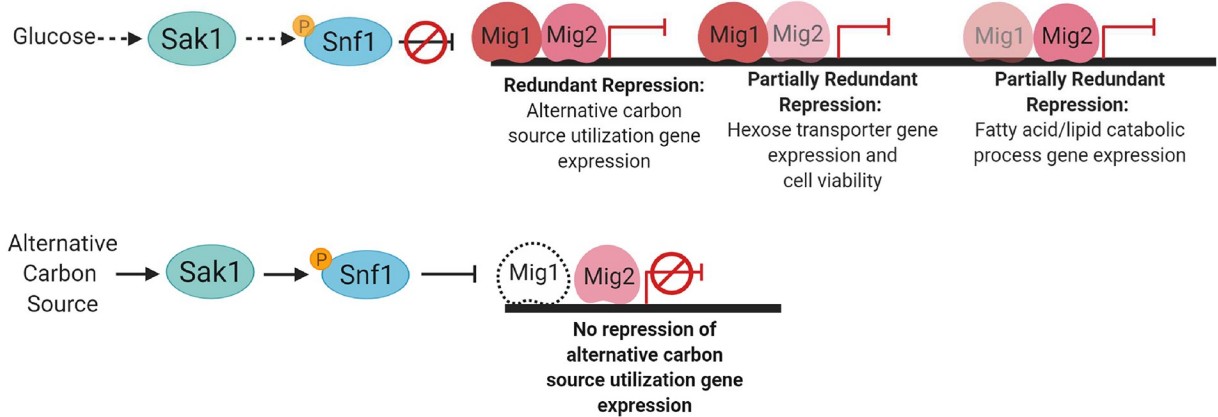

**Fig 8. Summary diagram of the roles of Mig1 and Mig2 in the glucose repression pathway in *C. albicans*.** Sak1 and Snf1 connections drawn from published data by Ramirez-Zavala et al. [9]. Black dotted arrows indicate unclear regulatory connections. Dotted Mig1 indicates reduced protein levels in cells grown on an alternative carbon source. No Mig2 protein level data exist to our knowledge.

the YEP maltose vs YEP glucose comparison [20]; another 43 of the genes are up-regulated in the YEP lactate vs 0.01, 0.1, or 1% glucose datasets [53]. The remaining genes are still enriched for oxidation-reduction and peroxisomal functions. It is possible that Mig1 and Mig2 respond to some signal in addition to presence of glucose, though the functional enrichment among these target genes suggests that the signal is related to carbon metabolism.

Some genes respond only in the *mig1Δ/Δ* or *mig2Δ/Δ* single mutants, and not in the *mig1Δ/Δ mig2Δ/Δ* double mutant. Most of these genes are down-regulated in the *mig* mutants. Because Mig1 and Mig2 seem to function primarily as repressors, we infer that down-regulated genes respond indirectly to Mig1/2 through compensatory responses to altered metabolism or other biological processes. About 10% of up-regulated genes respond only in one single mutant, and we suggest that they reflect compensatory responses as well.

## Mig1 and Mig2 regulation

In *S. cerevisiae*, *ScMIG1* and *ScMIG2* RNA levels vary with carbon source [13, 15]. In addition, *Sc*Mig1 is regulated at the level of nuclear translocation [17], whereas *Sc*Mig2 is regulated through protein degradation [16]. In *C. albicans* we found no evidence that RNA levels vary for *MIG1* or *MIG2*, in agreement with prior *MIG1* studies [18]. However, our data show that Mig1 protein accumulates at higher levels in glucose media than in non-repressing glycerol or mannitol media. It is possible that changes in Mig1 accumulation reflect carbon-responsive differences in its degradation rate, though this mechanism is not proven.

## Connection between Mig1/2 function and pathogenicity

A *mig1Δ/Δ mig2Δ/Δ* double mutant expresses many metabolic genes that are induced upon interaction with host cells. On that basis we anticipated that the double mutant would be "pre-adapted" to growth after uptake into macrophages and endothelial cells, and might then cause excessive damage to host cells. We observed the opposite result though; the *mig1Δ/Δ mig2Δ/Δ* double mutant caused reduced damage. One explanation is that metabolic flexibility is critical for effective host cell interaction, and the fact that the *mig1Δ/Δ mig2Δ/Δ* double mutant is locked in the glucose-derepressed state impedes interaction, perhaps at an early time when glucose is available. A second possibility is that the double mutant's cell wall integrity defect impairs its host cell interaction. Finally, we observed that the *mig1Δ/Δ mig2Δ/Δ* double mutant has impaired filamentation, a phenotype closely tied to host interaction. Sak1, Snf1, and now Mig1-Mig2 have all been connected to filamentation [9, 24, 25], though the mechanisms underlying this connection is unclear. Two major activators of filamentation, *BRG1* and *UME6*, are down-regulated in the *mig1Δ/Δ mig2Δ/Δ* double mutant during growth in YPG (S1 Table). A simple possibility is that levels of Brg1 and Ume6 mediate effects of Mig1/2 on filamentation.

## Placement of Mig1 and Mig2 in the Snf1 pathway

Conservation of the Snf1 pathway and the prior connection of Mig1 to glucose-repressed genes has suggested that Mig1 acts downstream of Snf1 [4], and these same arguments pertain to Mig2. In addition, here we have presented epistasis tests that support the hypothesis that Mig1 and Mig2 act downstream of Snf1 and Sak1. For example, *sak1Δ/Δ* mutant growth defects on YPG and Spider medium were partially suppressed by a *mig2Δ/Δ* mutation; suppression was more effective with a *mig1Δ/Δ mig2Δ/Δ* double mutation. Also, *snf1Δ/Δ* mutant inviability was relieved by a *mig1Δ/Δ* mutation to yield a phenotypically abnormal strain; phenotypic abnormalities were relieved by a *mig1Δ/Δ mig2Δ/Δ* double mutation. Nanostring gene expression readouts also supported this hypothesis. Specifically, *sak1Δ/Δ* gene expression

defects were partially reversed in either *sak1Δ/Δ mig1Δ/Δ* or *sak1Δ/Δ mig2Δ/Δ* strains, and a *sak1Δ/Δ* mutation had only minor gene expression impact in a *mig1Δ/Δ mig2Δ/Δ* double mutant background. Similarly, a *snf1Δ/Δ* mutation had only minor gene expression impact in a *mig1Δ/Δ mig2Δ/Δ* double mutant background. These observations are consistent with the hypothesis that Mig1 and Mig2 function downstream of Sak1 and Snf1 to mediate effects on gene expression and biological phenotype.

## Mig1 and the nature of Snf1 essentiality

The requirement of Snf1 for *C. albicans* viability [9, 21–24] can be explained by two hypotheses, first proposed by Kwon-Chung and colleagues [21]. Snf1 may be essential because of its function in the glucose repression pathway; for example, essential genes may be under glucose repression in *C. albicans*. Alternatively, Snf1 may be essential because it has a novel second role in *C. albicans* in addition to its role in glucose repression pathway, and that second role is essential for viability. The fact that loss of Mig1 permits recover of viable *snf1Δ/Δ* mutants provides strong support for the first hypothesis. Moreover, the finding helps refine the hypothesis in providing candidate genes that may be the essential targets of glucose repression. Given that *snf1Δ/Δ* mutants could be recovered in a *mig1Δ/Δ* background and not in a *mig2Δ/Δ* background, we infer that repression of Mig1-selective targets is the cause of *snf1Δ/Δ* mutant inviability in an otherwise wild-type background. Based on our analysis, there are thus only 67 candidate genes (Fig 2A and S1 Table). Although many Mig1-selective target genes are known to be nonessential, we suspect that reduced expression of multiple Mig1-selective targets may have additive effects that impact viability. One simple hypothesis is that *snf1Δ/Δ* mutant inviability is the result of a profound glucose uptake defect, given that glucose transporter genes are major Mig1-selective targets.

## Methods

### Media and culture conditions

Frozen strains were maintained in 15% glycerol frozen stocks at -80˚C. Streaked strains were maintained on YPD agar plates (2% dextrose, 2% Bacto peptone, 1% yeast extract, 2% Bacto Agar) and overnight cultures were grown in liquid YPD media (2% dextrose, 2% Bacto peptone, 1% yeast extract) rotating at 75 rpm at 30˚C in 15 ml culture tubes. Transformants were selected on synthetic media plates (2% dextrose, 1.7% Difco yeast nitrogen base with ammonium sulfate and necessary auxotrophic supplements) or selected for nourseothricin-resistance on YPD + 400 µg/ml nourseothricin (clonNAT, Gold Biotechnology). For alternative carbon source phenotyping, strains were spotted on YPG Plates (2% glycerol, 2% Bacto peptone, 1% yeast extract, 2% Bacto Agar) and Spider Medium Plates (1% Difco Nutrient Broth, 1% D-Mannitol, 0.2% Potassium Phosphate Dibasic ($K_2HPO_4$), 2% Bacto Agar). For transcript profiling, strains were grown in filter sterilized, liquid YPD, YPG, or Spider Medium. For hyphal growth and macrophage assays, strains were grown in filter sterilized Roswell Park Memorial Institute medium 1640 media (RPMI) pH 7.4.

### Strain construction

PCR products or linearized plasmids were transformed into *Candida albicans* cells using the lithium acetate transformation method [54]. Homozygous mutants were constructed in the SN152 or SN250 background strains [55] using the pSN69 (*Candida dubliniensis ARG4*), pSN52 (*Candida dubliniensis HIS1*), pSN40 (*Candida maltosa LEU2*), or p*NAT* [33] deletion cassettes. Homozygous mutants were created using a sgRNA targeting the gene of interest, a

repair DNA template containing a selection marker, and a transient CRISPR-Cas9 system as previously described in detail [33].

For western blotting analysis, a hemagglutinin (HA) epitope-tagged Mig1 strain was constructed using strain SN152. Primers KL44 and KL45 were used to amplify the HA tag and *HIS1* selection marker from plasmid pED3-HA (S2 Text) and contained homology to the 3' end of the *MIG1* coding region. A single guide expression cassette targeting the genomic region downstream of the *MIG1* coding region was created using primers KL46 and KL47. The transformation mix contained the DNA amplified from pED3-HA, Cas9, and the single guide expression cassette. Transformants were selected on CSM media lacking Histidine. To construct a prototrophic strain a single guide targeting the junction between the *leu2* upstream sequence and the deletion scar was created using primers KL48 and KL49 and *CmLEU2* and *CdHIS1* amplified from SN250 genomic DNA. The transformation mix contained the single guide expression cassette, Cas9, *CmLEU2*, and *CdHIS1* DNA. Transformants were selected on CSM media lacking Leucine and Histidine.

For a detailed description of strain construction, see Supplemental file S1 Text.

For a complete strain and primer list, see Supplemental files S7 and S8 Tables.

## Spotting plate assays

Strains were diluted in $H_2O$ to an $OD_{600}$ of 3.0 measured with a spectrophotometer. Five-fold dilutions were spotted using a multichannel pipette on the indicated media. Plates were incubated for 1–3 days at 37°C or 30°C as indicated.

## Hyphal microscopy and quantification

Cells from an overnight culture were diluted to an $OD_{600}$ of 0.5 in glass culture tubes containing 5 ml pre-warmed RPMI media. Cultures were grown rotating at 60 rpm at 37°C for 4 h. Cells were fixed with 4% formaldehyde and stained with Calcofluor white. Cells were visualized with a Zeiss Axio Observer Z.1 fluorescence microscope and a 20x objective.

For quantification of hyphal length, 10 fields of view were acquired for each strain analyzed using a 20x DIC objective with 1.6x optical zoom. Cells were measured from the beginning yeast cell to the end hyphal tip using the segmented line tool from ImageJ (https://imagej.nih.gov/ij/). Only cells where the entire hyphal cell was clearly visible in the field of view were measured. Significance was calculated using the GraphPad PRISM one-way ANOVA test followed by Tukey post hoc analysis, $p < 0.05$ significance.

## RNA-sequencing

RNA-sequencing and data analysis were performed as previously described [56] 5 μg of extracted RNA was treated with 2 units of TurboDNAse (Invitrogen) in a 50 μl reaction. To inactivate the DNAse, NaCl, Tris-HCl pH7.5, and EDTA, were added to a final concentration of 500 mM, 200 mM, and 10 mM respectively. The RNA was PCI extracted and the supernatant containing the RNA was purified over a Zymo Research RNA clean up column and eluted with 15 μl of nuclease free water. 2 μg of total RNA was used as input for the Lexogen mRNA sense kit v2. The kit was used according to the manufacturer's instructions for shorter amplicons. 11 PCR cycles were performed, and the libraries were run on a D1000 DNA tape (Tapestation) to assess the size and quality of the library. The concentrations of the libraries were measuring using the High sensitivity DNA assay for the Qubit (Invitrogen) The libraries were diluted to 8 nM and subjected to one lane of Illumina sequencing (Novogene), resulting in an average of 16 million reads per library.

Raw fastq reads were trimmed using cutadapt (v 1.9.1) with options "-m 42 -a AGATCG-GAAGAGC" to remove Illumina 3' adapter sequence and "-u 10 -u -6" to remove the Lexogen random priming sequences. Trimmed reads were mapped using tophat (v 2.0.8) [57] with options "–no-novel-juncs" and "-G" to align to the *C. albicans* SC5314 reference genome assembly 22. Primary alignments were selected using samtools (v 0.1.18) [58] with options "view -h -F 256". Gene counts were created using "coverageBed" from bedtools (v 2.17.0) [59] with option "-S" to count stranded alignments. RNA-Seq reads mapped to the two alleles of each gene were combined for further analysis. Differential expression was assessed using DEseq2 (v 1.18.1) [60] in RStudio (v 1.1.383) using default options (alpha = 0.05).

## Biofilm microscopy and quantification

Biofilms were prepared and imaged as previously described in detail [61]. Overnight cultures were diluted to an $OD_{600}$ of 0.2 in 2 ml RPMI media in 6 well culture plates containing 1.5 cm x 1.5 cm sized medical-grade silicone squares. After 90 minutes, the biofilm squares were dipped in sterile PBS to wash unadhered cells and placed in a new 6 well culture plate containing 2 ml of RPMI media. Biofilm cultures were incubated at 37°C with orbital shaking at 60 rpm. After 24 h, biofilms were fixed with 4% formaldehyde/2% glutaraldehyde for 20 minutes. Fixed biofilms were washed 2x with PBS and stained with Alexafluor 594-concanavalin A for 24 h at room temperature.

Stained biofilms were indexed matched by a series of dehydration steps followed by immersion in methyl salicylate (50:50 methanol:$H_2O$ 2x, 100% methanol 2x, 50:50 methanol:methyl salicylate 2x, and then 100% methyl salicylate2x). The index matched biofilm was viewed through a coverglass and imaged using a slit-scan confocal optical unit on a Zeiss Axiovert 200 microscope with a Zeiss 40x/0.85 NA oil immersion objective.130 images were obtained at a time using a focus increment of 0.9 μm. Biofilm image stacks were processed using ImageJ (https://imagej.nih.gov/ij/) for concatenation, background subtraction, reslicing, and max projection.

For biofilm depth quantification, 3 measurements from 3 biological replicates were measured using a slit-scan confocal optical unit on a Zeiss Axiovert 200 microscope with a Zeiss 40x/0.85 NA oil immersion objective. Significance was calculated using the GraphPad PRISM one-way ANOVA test followed by Tukey post hoc analysis, $p<0.05$ significance.

## Data analysis software

Venn diagrams analysis of gene expression data was performed using (http://bioinformatics.psb.ugent.be/webtools/Venn/). Area-proportional venn diagram images were created using the eulerr package in R. Statistical analyses were performed using GraphPad Prism version 8.00 (Graphpad Software, Inc., La Jolla). Heatmaps were created using MultiExperiment Viewer (MeV). Caspofungin minimum inhibitory concentration heatmap was created using JavaTreeView.

## Macrophage cytotoxicity

The macrophage cytotoxicity protocol was performed as described previously [62]. The J774A.1 murine macrophage cell line was graciously provided by Sai Gopalakrishna Yerneni from the lab of Dr. Phil Campbell. Cells were maintained in RPMI media without phenol red, with 10% Serum and 5% penicillin/streptomycin at 37°C in 5% $CO_2$. Cells were used from passages 8–16. 100 μl of macrophage suspension was plated at a concentration of $2.5 \times 10^5$ cells/ml overnight in a 96 well tissue culture treated polystyrene plate.

The following day, overnight cultures of *C. albicans* were subcultured in YPD media for 5 h. Subcultured cells were washed twice in PBS and diluted to a concentration of $3 \times 10^6$ in pre-warmed RPMI media without FBS, without phenol red, with 5% penicillin streptomycin at 37°C. Media was removed from overnight macrophage cultures and replaced with 150 µl of RPMI without FBS. 50 µl of *C. albicans* cells were added to each well for a MOI of 3. (3 *C. albicans* cells:1 macrophage). 6 wells of macrophages were not incubated with *C. albicans* cells for 3 spontaneous release control wells and 3 max release control wells. Cells were incubated for 5 h. Following incubation, the Pierce LDH Cytotoxicity Assay kit was used according to the manufacturer's instructions. To achieve max lactate dehydrogenase release, 10 µl lysis solution was added to each max release control well. For positive control wells, 200 µl of a 10 mL PBS + 1% BSA freshly made stock was added to 3 blank wells and 2 µl of positive control mix from the kit was added. Supernatant from all wells was diluted 1:5 in PBS and 100 µl was pipetted into a new 96 well plate. 50 µl of substrate mix was added to each well and incubated for 30 min. at RT. Following incubation, 50 µl of stop solution was added to each well.

Absorbance was read on a Tecan at 490 nm and background absorbance was read at 680 nm. The background absorbance was subtracted from the 490 nm reading. Percent cytotoxicity was calculated according to the manufacturer's guidelines. The assay was performed in triplicate to calculate percent cytotoxicity. Significance was calculated using the GraphPad PRISM one-way ANOVA test $p < 0.05$ significance.

## RNA extraction

For RNA-sequencing experiment and Nanostring experiment for Mig1/Mig2 strains, overnight cultures of cells were diluted to an $OD_{600}$ of 0.2 in 25 ml filtered YPD media. Cells were grown for 4 h shaking at 225 rpm at 37°C.

For Nanostring experiment of Sak1/Snf1 strains, overnight cultures of cells were washed in $H_2O$ and diluted to an $OD_{600}$ of 0.4 in 25 ml filtered YPG media. Cells were grown for 4 h shaking at 225 rpm at 37°C.

Cells were harvested via vacuum filtration and frozen at -80°C. RNA extraction was performed using Qiagen RNeasy Mini Kit (cat#74104) with modifications. Cells were washed from the membrane and resuspended with 1 ml of cold $H_2O$. The cell suspension was centrifuged at top speed for 30 sec. The supernatant was removed, and the cell pellet was resuspended with 600 µl of RLT + 1% *β*-Mercaptoethanol. The cell suspension was transferred to a screw cap tube containing 300 µl Zirconia beads (Fisher Scientific) and 600 µl phenol:chloroform:isoamyl alcohol 25:24:1. Tubes were vortexed using a mini-beadbeater (Biospec Products) for 3 min, and centrifuged at top speed for 5 min at 4°C. 550 µl of the top aqueous phase was transferred to a new screw cap tube containing 550 µl of 70% ethanol. The mixture was transferred to a RNeasy Mini Spin Column and RNA isolation was followed according to the manufacturer's guidelines.

## Nanostring

Nanostring analysis was performed as previously described [25]. Gene expression was measured using the nCounter SPRINT Profiler. For our analysis, 28 target genes and 5 normalization genes (*ARP3*, *FKH2*, *GIN4*, *TUP1*, and *CDC28*) were selected for the codeset. For each Nanostring assay, 100 ng of RNA was added to the Nanostring codeset mix and incubated at 65°C overnight (16–18 h). The samples were loaded onto the cartridge according to the manufacturer's instructions and placed in the instrument for scanning and data collection.

Heatmaps of Nanostring gene expression data represent were created using MultiExperimentViewer v4.8.1 (MeV) software. Colors represent log2-transformed ratios of gene expression comparisons. Hierarchical clustering was performed using average linkage clustering based on Manhattan distance and optimized for gene leaf order.

### Endothelial cell damage assay

Endothelial cell damage by *C. albicans* cells was assessed as previously described [63] by measuring $^{51}$Cr release. Human endothelial cells were maintained in M199 medium as previously described [63] and loaded with 5 μCi/ml $Na_2$ $^{51}CrO_4$ overnight. Cells were washed, and inoculated with *C.albicans* cells at a concentration of $4 \times 10^4$ organisms per well. Cells were incubated for 3 h and the $^{51}$Cr release was quantified using the formula: (experimental release—spontaneous release)/(total incorporation—spontaneous release). The assay was performed in triplicate using three biological replicates. Statistical significance was calculated using GraphPad Prism one-way ANOVA test $p < 0.05$ significance.

### Western blotting

Total soluble protein was extracted for Western blotting analysis from strain KL1026 and strain CW542 (for untagged negative control) in 50 ml cultures of the indicated media at 37°C, shaking at 225 rpm for 4 h. Cells were harvested by centrifugation and resuspended in FA lysis buffer (50 mM HEPES-KOH, 140 mM NaCl, 1 mM EDTA, 1% Triton X-100, 0.1% Sodium deoxycholate, 1 mM PMSF, and 1x proteinase inhibitor). Resuspended cells were homogenized by bead-beating with glass beads. Protein concentration was measured using a Bradford Assay(Bio-Rad). 30 μg of protein was loaded for each gel.

Protein extracts were separated on an 8% SDS polyacrylamide gel and transferred to PVDF membranes (Bio-Rad Laboratories). Ponceau S solution (Sigma) was used to assess equal loading. Membranes were blocked for 16 h at 4°C in 5% milk (Blotting-grade Nonfat dry milk, Bio-Rad Laboratories) in TBST (tris-buffered saline containing 0.05% Tween 20, pH 7.4). After washing in TBST, membranes were incubated with anti-HA monoclonal antibody (Roche; #11 583 816 001) at a 1:6,000 dilution for 1 h RT, washed with TBST, and then incubated with the secondary antibody, anti-mouse IgG-HRP(Santa Cruz Biotechnology; sc-516102), at a 1:5,000 dilution for 1 h RT. Signal was detected by enhanced chemiluminescence (Thermo Fisher Scientific, #32106) For loading control, the membranes were stripped for 10 min. using buffer containing 0.2 M glycine, 0.1%SDS, and 1% Tween 20 adjusted to pH 2.2. Membranes were washed in PBS and TBST before blocking again with 5% milk. The steps above were repeated for the stripped membrane except incubating with anti-tubulin rat monoclonal antibody(Abcam, #ab6046) at 1:6,000 dilution and Goat anti-Rat IgG HRP(Abcam #ab6734) at 1:5,000 dilution for the secondary antibody.

Blots were imaged using ChemiDoc Touch Imaging System (Bio-Rad Laboratories). Signal was quantified by densitometric analysis using FIJI (https://fiji.sc/). Mig1-HA signal was normalized to the Tubulin signal among the samples within the same blot.

### Supporting information

**S1 Fig. Intersections of YPG vs YPD responsive genes and related datasets.** Venn Diagram includes upregulated genes from *mig1Δ/Δ mig2Δ/Δ* vs wild-type YPD, wild-type YPG vs wild-type YPD, wild-type engulfed by macrophages vs non-engulfed [28], pH4 vs YPD [29], and biofilm cells vs yeast cells [30]. A list of the 116 shared genes can be found in tab S1G of S1 Table. The gene set is enriched for the GO terms "monocarboxylic acid catabolic process,"

"fatty acid catabolic process," and related terms (tab S2D of S2 Table).
(PDF)

**S2 Fig. Multiple sequence alignment of Mig proteins from *Saccharomyces cerevisiae* and *Candida albicans*.** Protein sequences were aligned using Clustal Omega and visualized using Jalview Clustal color scheme.
(PDF)

**S3 Fig. Western blot of Mig1-HA tagged protein grown in media with various carbon sources. A**. Western blots are from biological triplicates used to quantify Mig1-HA tagged protein levels. Protein extracts from strains KL1026 and CW542 were made from overnight cultures washed in $H_2O$, inoculated at OD 0.2, and grown in 50 ml cultures of the indicated medium for 4 h at 37˚C. For Spider + Glucose medium, 5 ml of 20% glucose was added to a culture grown in Spider medium at 3 h and 50 min. CW542 protein extracts were used as a non-tagged negative control. Ponceau S was used as a control for proper protein transfer. Samples were visualized with an antibody against HA and membranes were stripped and reprobed with an antibody against Tubulin for loading control. **B**. Functional phenotyping of Mig1 protein in the Mig1-HA strain using a caspofungin sensitivity plate dilution assay from Fig 6A. The Mig1-HA strain is not more sensitive to caspofungin compared to the wild-type.
(PDF)

**S4 Fig. Independent isolates of *sak1Δ/Δ mig1Δ/Δ* and *sak1Δ/Δ mig2Δ/Δ* strains show similar growth phenotypes on Spider media.** Strains: Wild-type (CW542), *sak1Δ/Δ* (KL988), *sak1Δ/Δ mig1Δ/Δ* (KL951 and KL952), *sak1Δ/Δ mig1Δ/Δ mig2Δ/Δ* (KL955) and *sak1Δ/Δ mig2Δ/Δ* (KL960 and KL962) were grown overnight in YPD and tenfold serial dilutions of the indicated strains were spotted on YPD and Spider plates. Growth was visualized after 2 days of incubation at 37˚C.
(PDF)

**S5 Fig. The *snf1Δ/Δ mig1Δ/Δ* strain shows abnormal filamentation and coloration compared to wild-type and the *snf1Δ/Δ mig1Δ/Δ mig2Δ/Δ* strain.** Strains: Wild-type (CW542), *snf1Δ/Δ mig1Δ/Δ* (KL953 and KL954), *snf1Δ/Δ mig1Δ/Δ mig2Δ/Δ* (KL957 and KL958) and *snf1Δ/Δ mig1Δ/Δ mig2Δ/Δ + SNF1* (KL974) were spotted at an OD of 0.1 on YPD media and grown at 30˚C for 7 days.
(PDF)

**S1 Text. Strain construction.** Details of strain constructions are provided.
(DOCX)

**S2 Text. Plasmid ED3-HA sequence.** The sequence of plasmid ED3-HA is provided.
(DOCX)

**S1 Table. RNA-seq.** RNA-seq data are provided.
(XLSX)

**S2 Table. GO terms.** GO terms associated with gene subsets are provided.
(XLSX)

**S3 Table. FET YPG vs YPD.** Comparisons of datasets via Fisher's Exact Test are provided.
(XLSX)

**S4 Table. NanoString complementation.** Gene expression data for mutant and complemented strains are provided.
(XLSX)

**S5 Table. NanoString spider medium.** Gene expression data for cells grown in Spider medium are provided.
(XLSX)

**S6 Table. NanoString carbon codeset.** Detailed gene expression data underlying Fig 5 are provided.
(XLSX)

**S7 Table. Primer list.** Sequences of primers used in this study are provided.
(XLSX)

**S8 Table. Strain list.** Genotypes of strains used in this study are provided.
(XLSX)

# Acknowledgments

We are grateful to Dr. Fred Lanni, Dr. Eunsoo Do, and Max Cravener for discussions and technical assistance and Saigopalakrishna Yerneni for graciously providing the J774A.1 cells used in this study.

# Author Contributions

**Conceptualization:** Katherine Lagree, Aaron P. Mitchell.

**Data curation:** Katherine Lagree, Carol A. Woolford.

**Formal analysis:** Katherine Lagree, Carol A. Woolford, Manning Y. Huang, C. Joel McManus, Scott G. Filler.

**Funding acquisition:** Scott G. Filler, Aaron P. Mitchell.

**Investigation:** Katherine Lagree, Manning Y. Huang, Gemma May, Norma V. Solis.

**Methodology:** Carol A. Woolford, Manning Y. Huang, Gemma May, C. Joel McManus, Norma V. Solis.

**Project administration:** Aaron P. Mitchell.

**Resources:** Aaron P. Mitchell.

**Supervision:** C. Joel McManus, Scott G. Filler, Aaron P. Mitchell.

**Writing – original draft:** Katherine Lagree.

**Writing – review & editing:** Katherine Lagree, Aaron P. Mitchell.

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
