## [Decision Letter · Decision Letter 0]

10 Sep 2019

Dear Aaron,

Thank you very much for submitting your Research Article entitled 'Resolving the roles of Mig1 and Mig2 in the Sak1/Snf1 controlled glucose repression pathway in Candida albicans' to PLOS Genetics. Your manuscript was fully evaluated at the editorial level and by three independent peer reviewers all of whom work on Candida as the experimental system (fungal genetics, molecular medical mycology). The reviewers appreciated the attention to an important problem, but raised some concerns about the current manuscript. Based on the reviews, we will not be able to accept this version of the manuscript, but will be happy to review again a revised version (this constitutes major revision with respect to two reviews, and minor revision with respect to the other). As journal policy, we cannot, of course, promise publication at that time.

Your revisions should address the specific points made by each reviewer. We also ask that you include a detailed list of your responses to the review comments and a description of the changes made in the manuscript.

In revising the manuscript for further consideration at PLOS Genetics, please aim to resubmit within the next 60 days, unless it will take extra time to address the concerns of the reviewers, in which case we would appreciate an expected resubmission date by email to plosgenetics@plos.org.  The revised manuscript will be sent to one or more of the original reviewers, depending on revisions and availability of the reviewers.

[LINK]

We are sorry that we cannot be more positive about your manuscript at this stage. Please do not hesitate to contact us if you have any concerns or questions.   We thank you for your support of the Public Library of Science in General, and PLOS Genetics in particular.

Yours sincerely,

Joseph Heitman, MD, PhD

Associate Editor

PLOS Genetics

Gregory P. Copenhaver

Editor-in-Chief

PLOS Genetics

Reviewer's Responses to Questions

**Comments to the Authors:**

Reviewer #1: The manuscript by Mitchell et al describes the role of Mig1 and Mig2, two repressors that repress genes involved in carbon source utilization and catabolite repression. The authors show that Sak1 is required for growth in the absence of glucose, and that Mig1 and Mig2 are epistatic to Sak1, in a partially redundant way. The paper begins with a transcriptomics analysis of cells grown in the presence of either glucose of glycerol and includes mig1, mig2, or mig1mig2 mutants, continues to nicely phenotype double and triple mutants along with complemented derivatives to support their genetic model, then characterizes the role of Mig1 and Mig2 in several important phenotypes. The manuscript is well-written and tells a nice story, and provides a satisfying explanation for the essentiality of Snf1 in C. albicans. I have a few comments that might strengthen the manuscript.

1) The authors points on the relatedness between their data set and other datasets could be more useful with more detail on whether the same genes (a subset) drove the similarity to the differentially expressed genes in other datasets. For example, were similar subsets of genes among those that drove similarity to macrophage experiments and M199-YPD comparisons? Did the Mig1 and Mig2 and Mig1/2 regulons coincide with any of these experiments? The comparison to other datasets might be useful later on in the paper once the Mig regulons have been defined. As written, this part of the paper is rather speculative.

2) On p. 14, the following text is found: “Growth properties of representative snf1Δ/Δ derivatives of mig1Δ/Δ and mig1Δ/Δ mig2Δ/Δ strains suggested that loss of Mig1 does not bypass the need for Snf1 entirely. For example, snf1Δ/Δ mig1Δ/Δ colonies exhibited hyperfilamentation and a peculiar yellow color, whereas snf1Δ/Δ mig1Δ/Δ mig2Δ/Δ colonies did not (Fig S1).”

The authors might consider whether the yellow color is riboflavin. Several studies have shown that metabolic perturbation and alterations in cAMP signaling can change riboflavin production. Here is one paper that references riboflavin production but it is not the first (PMID:21992526). Shining a UV light on the colony may reveal characteristic fluorescence and a simple extraction can be analyzed for the reduced and oxidized spectra characteristic of this molecule.

3) Fig. 4 should be revised to have genes in all capitals in the right-hand column and the text resolution is of a lesser quality than for other figures.

4) Italics needed in at least one figure legend and references need to be joined within a single bracket in the introduction.

5) Could the authors expand on the statement that “peroxisome biogenesis [is a] functions that align well with alternative carbon source utilization”.

6) Do the transcriptomics data still need to be deposited?

Reviewer #2: Regulation of sugar usage is very important for cells, and investigating the mechanisms by which organisms control which sugars are metabolised is a significant scientific endeavor. In the manuscript the Mitchell and Filler labs investigate the roles of Mig1 and Mig2 in glucose repression in the fungal pathogen C. albicans. The basic strategy is to make single and double mutants, and investigate the phenotypes in terms of sugar usage (glucose vs alternative sources) and, because the organism is a pathogen, the consequences for pathogenicity and traits associated with this lifestyle.

In the well-investigated model yeast S. cerevisiae, glucose repression has been connected to the Snf1 kinase and its regulation of the key transcription factor Mig1 – Mig1 serves as a repressor of alternative carbon sources. High glucose, inactive Snf1, active Mig1, and repression of the use of alternative sugars; low glucose, active Snf1, inactive repressor Mig1, and alternate sugars can be used. There are lots of nuances and other players in the circuitry, but the Snf1/Mig1 interaction is central to the control process. However, S. cerevisiae is somewhat of an outlier in aspects of carbon utilization relative to other fungi, in particular its propensity to ferment glucose even in the presence of oxygen (the Crabtree/Warburg effect), so it is reasonable to ask if the glucose circuitry is also controlled by Snf1/Mig1 in a non-Crabtree yeast like C. albicans.

Basically, the results of this study say yes – the Snf/Mig regulatory circuit controls alternative carbon use in C. albicans.

First, the authors do an RNAseq analysis of glucose vs glycerol grown cells and identify gene sets that are up-regulated (~ 400) and down-regulated (~ 100) in response to the growth in glycerol. These genes are compared to the expected genes from studies in S. cerevisiae, and to data sets for other comparisons in C. albicans, and are found to represent a sensible data set. Thus C. albicans appears to have a “standard” glucose repression response.

Next, they examine whether Mig1 and Mig2, two paralogs of the Mig1/2 repressors in S. cerevisiae, are important for glucose repression. The phylogenetic relationships of Mig1/2 in S. cerevisiae and Mig1/2 in C. albicans needs to be better explained. Essentially, as C. albicans is a pre WGD yeast, and S. cerevisiae is a post WGD yeast, there are several possible ways you could get a Mig1 and a Mig2 in both species. It is worth looking to see if a prediction can be made about whether the pre WGD yeast had a Mig1 and a Mig2 that became Mig1A and Mig1B; Mig2A and Mig2B after the duplication and resolved back to Mig1A and Mig2B, or did it resolve to Mig1A, Mig1B in yeast and lose the whole Mig2 lineage from before the WGD? Lots of possible arrangements, worth seeing if something could be said, especially given that the Mig1 and Mig2 in C. albicans seem to be functionally closer to each other than the Mig1 and Mig2 of S. cerevisiae.

To investigate the functions of the two candidate repressors, they make the single and double mutants, complement the genes by dropping versions in at the standard RPS1 locus, and, in the cases where position effects prevent full complementation, they replace to WT gene at the WT locus. This provides confidence the phenotypic assessments to follow are really the result of the loss of Mig1 or Mig2 function.

The first set of phenotypes is to see which genes are de-repressed in the absence of either or both the repressors. The authors identify genes that are repressed by both (need to be mig1-mig2- to see de-repression), repressed specifically by Mig1 (de-repressed in the mig1- or the mig1-mig2- strains), and repressed specifically by Mig2. The suggest that the specifically repressed genes can be functionally sorted – Mig2 for glyoxylate and beta oxidation, Mig1 for transporters, Mig1 and Mig2 for everything else, and in fact the majority of the regulated genes. Finally, they argue the Mig1/2 regulated genes are the same genes that show glucose repression, and that glucose repression of these genes doesn’t work well in the double mutant. These data sets do not correlate perfectly of course, and the failure of glucose repressed genes to respond in the mig1/2 double mutant is not absolute (still some apparent repression of some genes), but the patterns seen strongly support the central premise here.

Next the authors address the relationship of Snf1 and the Mig1/2 elements. Loss of Snf1 is lethal to C. albicans cells, this is somewhat surprising and has greatly limited analysis of the element. Based on the hypothesis that over-suppression of the Mig gene targets could be the cause of the lethality, the authors attempted to rescue the snf1 nulls be deleting the repressors, and were able to get viable snf1 nulls in the mig1- and mig1-mig2- backgrounds. Thus, overactivity of Mig1, presumably suppression of some essential function(s), appears to be the cause of the Snf1 null’s inviability. They follow this up by assessing the impact of the mig mutants on the regulatory kinase upstream of Snf1, and show that growth defects associated with loss of this kinase are abrogated by the repressor deletions, but find that the mig2 repressor is more important, in contrast to mig1 which is the cause of the snf1 inviability.

The authors then use nanostring probes for a set of genes defined as mig1, mig2 or mig1/mig2 responsive, and probe the consequences of various perturbations. Implementing the Sak1 response needs both mig1 and mig2, but there are genes that seem to specifically require one or the other. Snf1 also seems to go through mig1/2, with some specific differences from the Sak1 data set.

Other phenotypes are probed – response to wall stress, ability to make hyphae, to interact with immune and endothelial cells. Mig1/2 function is needed, either gene alone does a pretty good job, so the proteins appear close to redundant for these functions. The data in general are convincing and sensibly interpreted (with the caveats noted below), and the results suggest logical explanations for otherwise somewhat surprising results such as the difference in snf1 and sak1 phenotypes and the essentiality of snf1.

My major concern with this work is the context in which it is embedded. At one level, the sak1, snf1, mig1 circuit controls glucose repression in C. albicans essentially as it does in S. cerevisiae, so nothing fundamental has been changed in or added to our understanding of the process. To justify the work to the audience of PLoS Genetics the authors want to note the way C. albicans has structured the circuit relative to S. cerevisiae – to emphasize the differences. But here they seem to somewhat lose their way. The fact that in C. albicans the paralogs Mig1 and Mig2 act similarly, but have a selective bias in targets is stated as surprising, when it is exactly what should happen with paralogs in this situation; they diverge in function, but modestly. The suggestion that the trajectory of Mig function “evolution” is that it has expanded in C. albicans relative to S. cerevisiae seems less likely than that the arrangement in the post WGD yeast S. cerevisiae has compacted relative to the pre WGD yeast C. albicans. If the authors are going down this route they should be providing genomic evidence from phylogenetically intermediary species, and showing the real relationships among the Mig orthologs, as previously noted.

Minor points

The authors have left PMID identifiers instead of actual references in a few places.

Reviewer #3: This work addresses the SNF1/ AMP-activated protein kinase signaling system in C. albicans. RNA seq was used to probe the relationship between the transcriptional regulators of carbon source utilization, Mig1 and Mig2, in Candida albicans, and their upstream regulators, the kinases Snf1 and Sak1. Deletion mutants in the genes encoding these regulators were created singly and in combination with each other, and the effects of the mutations on the transcriptional profile of mutants was examined. Nanostring analysis of 28 genes was used to further examine the effects of these mutations on gene expression. Growth, filamentation and biofilm phenotypes of the mutants were analyzed. Lastly, the behavior of mig1 mig2 double mutants during the interaction with model host cells, macrophages and human umbilical vein endothelial cells, was tested.

The authors first established, by RNA seq, the genes differentially expressed in wild type cells growing in the preferred carbon source glucose, or an alternative carbon source, glycerol, in complex rich medium; they found an expected assortment of genes, the majority of which were upregulated. They next obtained RNA seq datasets for deletion mutants in MIG1, MIG2 and MIG1MIG2 grown in glucose, the carbon source in which the activity of these regulators is predicted to be most important, and again found a majority of genes with increased expression among the total of genes with altered expression levels. Presence of a binding motif for Mig1 (previously established by others) was examined for promoters of genes whose expression was altered in these mutants, and was found to occur at significant frequency in genes whose expression was increased, but not those whose expression was decreased in mig1, mig2 and mig1 mig2 mutants. (No further analysis was attempted of this observation, though an important consideration is whether these regulators control the expression of other transcriptional activators that are responsible for loss of expression of this set of genes.)

The investigators then turned to the specific roles played by Mig1 and Mig2. They found a set of genes whose expression levels were increased only when MIG1 was deleted. Among them were plasma membrane hexose transporters, HGT genes, of which 8 were expressed at increased levels. Genes whose expression increased only when MIG2 was deleted included genes required for lipid catabolism; hence these regulators have distinct specificities in the processes they target.

Most genes whose expression was increased in the mig1 mig2 mutant were unchanged in either of the single mutants; the authors concluded that Mig1 and Mig2 functions largely overlap.

Constructing double mutants in SNF1 with MIG1, MIG2 or MIG1 MIG2, the authors were able to recover snf1 mutants only in a background in which MIG1 was deleted. They then examined growth phenotypes of sak1 mutants and found that their growth defects were suppressed in mutants also lacking MIG2, and further suppressed in when MIG1 was additionally, but not solely, lacking.

The authors then used a panel of 28 nanostring probes to investigate the effects of mutating these regulators singly and in combination. They observed very small numbers of differentially regulated genes among that set. The mutants were then subjected to cell wall stress using caspofungin, to which mutants lacking Mig1 were hypersensitive. Filamentation and biofilm phenotypes of mig1mig2 double mutants were shown, and the same mutants were defective in damaging macrophages and endothelial cells.

The role of the orthologous genes and their products, in particular SNF1, as an ortholog of the mammalian AMP kinase, has been studied extensively in S. cerevisiae, and in other work in C. albicans. The study is a useful compilation of datasets and shows unique results specific to C. albicans, and as such makes a significant contribution to the field of Candida albicans physiology. Superficiality in experimental design and of the analysis, to some extent limits the potential of this manuscript to move our understanding forward.

Major issues:

The analysis is somewhat superficial at several points. E.g.:

1. Is it really an “overlap of the functions” of Mig1 and Mig2 that leads to increased expression of a majority of altered genes in the mig1mig2 double mutant, but not in either single mutant? Certainly redundant activities of these transcriptional regulators could explain this observation. Alternatively, a requirement for cooperative binding at the promoters of these genes, for mutual recruitment, for nucleosome placement etc. (“combinatorial cis-regulation”) (PMID: 19029883) is possible; the authors do not consider any alternative possibilities despite the burgeoning field investigating these regulatory mechanisms, in which Mig1 is a model. Instead they simply state, for genes whose increased expression in the absence of one of the regulators, is further increased when the other is also absent, “We infer for these genes that Mig1 or Mig2 is the major repressor, but that the other Mig1/2 protein can repress weakly in the absence of the first. Therefore, while Mig 1 nd Mig2 each have some specific target genes, they also have highly redundant roles in gene regulation.”

This a priori assumption of redundant functions of these transcriptional regulators, in light of today’s large literature on this subject - (E.g.: “Understanding these rules, which are often referred to as ‘cis-regulatory grammar’, is a major challenge in modern biology,” PMID: 30570483 or “discuss how different strategies, including extensive cooperative regulation (both direct and indirect), progressive priming of regulatory elements, and the integration of activities from multiple enhancers, confer specificity and robustness to transcriptional regulation during development” PMID: 22868264) - does not seem to be a sufficient consideration of the regulatory mechanism involved.

If the “overlap of functions” idea is correct, why is the expression of 14% of genes, whose expression is higher, in mig1 or mig2 single mutants or in both than in wild type, not higher in the mig1 mig2 mutant (38/268) (Fig. 2A)? Among the genes expressed at higher levels in the mutants than the wild type, this fraction amounts to fully one third (72/216)?

Similarly, the sentences starting with “The majority of genes that are up-regulated in the mig1D/D mig2D/D double mutant are not up-regulated …” and ending with “and we refer to them as Mig1/2 shared genes” seem very unclear at best.

2. As 8 HGT genes (though I count 10 in Fig. 2C?) are expressed at elevated levels in mig1 mutants growing in glucose, Mig1 seems to act as a repressor of these genes in glucose. Is it worth considering which hexoses may the products of these genes transport as apparently it is unlikely to be glucose? Or if it is glucose, why does Mig1 normally repress them while cells are growing in glucose?

3. Using a panel of only 28 nanostring probes seems insufficient to actually move the understanding of these regulators forward. In fact, only 24 are used, because 4 probes examine the expression of the genes under study, i.e. MIG1, MIG2, SNF1, SAK1. How were these 24 genes chosen? If one were to decide that only 24 genes are sufficient, would one not choose only effectors of the processes under study, e.g. enzymes, while avoiding regulators like the transcription factors TYE7 or ACE2? Having such a small sample of genes in the set, it is inaccurate to write “Nanostring analysis indicated that there may be fairly few Mig1-selective target genes expressed in a snf1D/D mig1D/D mutant.”

4. How can a statement that “Mutations in MIG1 and MIG2 are epistatic to mutations in SAK1 and SNF1” (Fig. 4 legend) be made when no mutants in MIG1 or MIG2 alone were tested? The only mutants tested in this figure are double and triple mig1 and mig2 mutants with sak1 or snf1 mutations. While for snf1, single mutations in this gene lead to inviability, this is not the case for sak1. With this experimental design, how can the authors claim that their observed results are not due e.g. to synthetic effects between the mutations? - From S. cerevisiae, it is known that Mig1 and Mig2 act downstream of Sak1 and Snf1 so the statement about epistasis seems very logical to the reader; but if the authors aim to be showing an original result and drawing a legitimate conclusion from their own finding, the experimental design must be such that this conclusion can be drawn. To remind the authors of “epistasis,” “For any pair of genes X and Y, we evaluated the level of epistasis by comparing the fitness WXY of the double mutant with the product of the fitness values WX and WY of the corresponding single mutants,” (PMID: 15592468)

5. Issues not considered experimentally e.g. other non-glucose carbon sources, either fermentable (e.g., fructose, galactose, maltose) or nonfermentable (e.g., oleic acid, since Mig2 seemed to target lipid catabolism genes) should at least have been considered in the discussion?

6. Would it not have contributed to the analysis to use a loss-of-function strain in SNF1 other than the homozygous deletion that was known to be inviable? (e.g. the heterozygote, a strain expressing a destabilized transcript as in PMID:31118301, etc. )

6. In S. cerevisiae, participation of the heterotrimeric SNF1 holoenzyme in multiple signaling pathways is long established (PMID: 17981722); the Snf1 catalytic subunit complexes not only with a gamma subunity Snf4 but with one of 3 possible beta subunits (Gal83, Sip1 or Sip2) which regulate its subcellular localization. Its regulation involves not just an upstream activating kinase like Sak1 (or Elm3 or Tok1), but also a specific phosphatase (regulatory subunit Reg1 and the catalytic subunit Glc7) and autoinhibition modulted by the gamma subunit. Among multiple regulatory mechanims including control of transcriptional regulators beyong Mig proteins, Snf1 controls the activity of metabolic enzymes by phosphorylation, such as Pfk27, Gpd2, and Acc1 (PMID: 25512067 ). SNF1 also is activated in response to stressors like alkaline stress and sodium stress, with differential responses: accumulation in the nucleus in response to alkaline but not sodium stress (PMID:17438333). Given the complexity already known from model yeast and other eukaryotes, positing simple linear models for C. albicans is not likely to optimize our understanding of the role of this important regulator. A linear model of signaling from Sak1 -> Snf1 -> Mig1 and Mig2 as in Fig. 7 clearly cannot adequately represent the findings. If the fungal cell required only linear signaling, presumably a two component system as in bacteria would suffice to respond to the availability of glucose. A recent paper comparing SNF1 signaling of Kluyveromyces lactis with that of S. cerevisiae shows how this might be done; they devised a less crude model of their findings (PMID: 26440109).

Minor issues:

1. A reviewer’s work is made more time consuming by the absence of line numbering in the manuscript.

2. “Carbon-source dependent transcriptional regulation of ScMIG1 and ScMIG2 expression has apparently been lost for C albicans MIG1 and MIG2.” Unless the authors have evidence that the ancestral trait is carbon-source dependent transcriptional regulation of the MIG genes, while constitutive expression of these regulators is a derived trait, it is inappropriate to refer to such an evolutionary relationship. They should simply state the observation.

3. This reviewer finds the expression “up-“ or “downregulated” to be somewhat inaccurate when describing the effect of mutations of transcriptional regulators on the expression of genes. Some of the effects of these mutations may be due to regulation, i.e. compensatory mechanisms of the cell to adapt to loss of a regulator; these effects are likely to be indirect. The most direct effects of the absence of these regulators, however, are aberrantly high or low expression levels of their target genes, i.e. elevated or diminished in comparison with the wild type.

4. Setting up a hypothesis that mig1 mig2 mutants might be more fit during phagocytosis by macrophages seems a bit contrived and is not required to justify this experiment. It would be highly surprising if loss of 2 important transcriptional regulators were to increase fitness in a model host interaction. These phenotype experiments with the mutants (caspofungin sensitivity, biofilm formation, interaction with macrophages and endothelial cells) are of interest and do not need artificial justification.

**Have all data underlying the figures and results presented in the manuscript been provided?**

Reviewer #1: No: Transcriptomics data may still need to be deposited.

Reviewer #2: Yes

Reviewer #3: Yes

PLOS authors have the option to publish the peer review history of their article (what does this mean?). If published, this will include your full peer review and any attached files.

Reviewer #1: No

Reviewer #2: No

Reviewer #3: No

---

## [Decision Letter · Decision Letter 1]

18 Dec 2019

Dear Aaron,

Thank you very much for submitting your revised Research Article entitled 'Roles of Candida albicans Mig1 and Mig2 in glucose repression, pathogenicity traits, and SNF1 essentiality' to PLOS Genetics. Your manuscript was fully evaluated at the editorial level and by the same three independent peer reviewers who reviewed the original submission. The reviewers appreciated the attention to an important topic.  All three reviewers recommended acceptance.  Reviewers 1 and 2 indicated that all of their comments raised had been addressed.  Reviewer 1 had one very minor comment about a supplemental table.  Reviewer 3 raised two minor remaining issues that may be considered stylistic, or may have something more to them.  In addition, this reviewer noted two minor issues that may require correction or editing.  We are therefore returning the manuscript to you as we expect that you may want to address these points raised in some fashion by judicious editing.  It is not our intent to then subject this to another round of review, as all three reviewers indicated that from their perspective the manuscript was acceptable.   

We ask that you:

1) Provide your responses to the review comments and a description of any changes you have made in finalizing the manuscript.

We hope to receive your revised manuscript within the next 7 days. If you anticipate any delay in its return, we would ask you to let us know the expected re-submission date by email to plosgenetics@plos.org.

[LINK]

Please let us know if you have any questions while making these revisions.  We very much appreciate your submitting this interesting study to PLOS Genetics, and look forward to receiving the finalized manuscript which will be accepted and published.

Yours sincerely,

Joe

Joseph Heitman, MD, PhD

Associate Editor

PLOS Genetics

Gregory P. Copenhaver

Editor-in-Chief

PLOS Genetics

Reviewer's Responses to Questions

**Comments to the Authors:**

Reviewer #1: The revised manuscript is a pleasure to read and represents a much more focused story that will be easier for readers to launch from. I have read through the paper and analyzed the new figures, and have no concerns.

The authors may consider a brief legend for supplemental tables to help with navigation within complex tables (e.g. Table S4).

Reviewer #2: Revision has dealt well with my concerns

Reviewer #3: The revised manuscript is improved and much clearer. Essentially it is a transcriptional analysis of the role of MIG1 and MIG2, which is worthwhile. It suggests that it is the inability to relieve otherwise constitutive transcriptional repression of Mig1 that makes snf1 mutants inviable.

This reviewer is still somewhat uncomfortable with the manuscript’s manner in setting up results as surprises or answers to mysteries or puzzles e.g.

- essentiality of SNF1; the original Petter et al report in 1997 discussed the possibility of Snf1 possessing “some other unknown cellular functions” unrelated to carbon source adaptation not as an explanation for their inability to isolate homozygous null mutants in this gene, but because of the heterozygotes’ phenotypes of wild type carbon source utilization patterns together with a significant decrease in growth rate as well as morphological changes.

- the finding of decreased virulence behaviors of mig1 mig2 mutants in a macrophage model, when metabolic flexibility has been described by many authors as a critical virulence determinant in C. albicans (e.g. Ramirez-Zaval et al 2017). The rationale presented by the authors in their response is not convincing to this reviewer - 3 out of the 4 important genes whose mutants they cite as showing increased virulence have not been specifically studied for this phenotype to my knowledge, and the 4th, PHO4, has decreased virulence in several in vivo models.

This is a matter of taste more than anything else and just a suggestion to the authors that their readers may appreciate a measured presentation of rationales for experiments.

Minor questions: why does the sak1 mutant appear twice in the heat map of Fig. 5?

Fig. 1B is mentioned in the text after Fig 2?

**Have all data underlying the figures and results presented in the manuscript been provided?**

Reviewer #1: Yes

Reviewer #2: Yes

Reviewer #3: Yes

PLOS authors have the option to publish the peer review history of their article (what does this mean?). If published, this will include your full peer review and any attached files.

Reviewer #1: No

Reviewer #2: No

Reviewer #3: No

---

## [Editor Report · Decision Letter 2]

20 Dec 2019

Dear Aaron,

We are pleased to inform you that your revised manuscript entitled "Roles of Candida albicans Mig1 and Mig2 in glucose repression, pathogenicity traits, and SNF1 essentiality" has been editorially accepted for publication in PLOS Genetics. Congratulations!  We very much appreciate your detailed and clear responses to reviews, and the attention to detail in finalizing this for publication. 

Thank you again for supporting open-access publishing; we are looking forward to publishing your work in PLOS Genetics!  All best for the holiday season ahead.

Yours sincerely,

Joe

Joseph Heitman, MD, PhD

Associate Editor

PLOS Genetics

Gregory P. Copenhaver

Editor-in-Chief

PLOS Genetics

Comments from the reviewers (if applicable):

**Data Deposition**

http://datadryad.org/submit?journalID=pgenetics&manu=PGENETICS-D-19-01346R2

**Press Queries**

---

## [Editor Report · Acceptance letter]

13 Jan 2020

PGENETICS-D-19-01346R2 

Roles of Candida albicans Mig1 and Mig2 in glucose repression, pathogenicity traits, and SNF1 essentiality 

Dear Dr Mitchell, 

We are pleased to inform you that your manuscript entitled "Roles of Candida albicans Mig1 and Mig2 in glucose repression, pathogenicity traits, and SNF1 essentiality" has been formally accepted for publication in PLOS Genetics! Your manuscript is now with our production department and you will be notified of the publication date in due course.

With kind regards,

Kaitlin Butler

PLOS Genetics

On behalf of:
